

# Evidence of small-scale quasi-isentropic mixing in ridges of extra-tropical baroclinic waves

Daniel Kunkel[1], Peter Hoor[1], Thorsten Kaluza[1], Jörn Ungermann[2], Björn Kluschat[1], Andreas Giez[3], Hans-Christoph Lachnitt[1], Martin Kaufmann[2,4], and Martin Riese[2,4]

[1]Institute for Atmospheric Physics, Johannes Gutenberg University, Mainz, Germany
[2]IEK-7, Forschungszentrum Jülich, Jülich, Germany
[3]Flight Experiments, Deutsches Zentrum für Luft- und Raumfahrt, Oberpfaffenhofen, Germany
[4]Institute for Atmospheric and Environmental Research, University of Wuppertal, Wuppertal, Germany

**Correspondence:** Daniel Kunkel (dkunkel@uni-mainz.de)

**Abstract.** Stratosphere-troposphere exchange within extratropical cyclones provides the potential for anthropogenic and natural surface emissions to rapidly reach the stratosphere as well as for ozone from the stratosphere to penetrate deep into the troposphere, even down into the boundary layer. The efficiency of this process directly influences the surface climate, the chemistry in the stratosphere, the chemical composition of the extratropical transition layer, and surface pollution levels. Here, we present evidence for a mixing process within extratropical cyclones which has gained only little attention so far and which fosters the transport of tropospheric air masses into the stratosphere in ridges of baroclinic waves. We analyzed airborne measurement data from a research flight of the WISE (Wave driven isentropic exchange) campaign over the North Atlantic in autumn 2017 supported by forecasts from a numerical weather prediction model and trajectory calculations. Further detailed process understanding is obtained from experiments of idealized baroclinic life cycles. The major outcome of this analysis is that air masses mix in the region of the tropopause and potentially enter the stratosphere in ridges of baroclinic waves at the anti-cyclonic side of jet without changing their potential temperature drastically. This quasi-isentropic exchange occurs above the outflow of warm conveyor belts, in regions which exhibit enhanced static stability in the lower stratosphere and a Kelvin-Helmholtz instability across the tropopause. The enhanced static stability is related to radiative cooling below the tropopause and the presence of small scale waves. The Kelvin-Helmholtz instability is related to vertical shear of the horizontal wind associated to small scale waves at the upper edge of the jet-stream. The instability leads to the occurrence of turbulence and consequent mixing of trace gases in the tropopause region. While the overall relevance of this process has yet to be assessed, it has the potential to significantly modify the chemical composition of the extratropical transition layer in the lowermost stratosphere in regions which have previously gained only little attention in terms of mixing in baroclinic waves.



# 1   Introduction

The extratropical transition layer (ExTL) as a unique feature of the extratropical upper troposphere and lower stratosphere (UTLS) is a direct consequence of the exchange and consequent mixing between air masses from the stratosphere and troposphere (e.g., Danielsen, 1968; Gettelman et al., 2011). The depth of this transition layer is commonly diagnosed from vertical

distributions of certain trace species with distinct tropospheric and stratospheric sources and from tracer-tracer correlations (e.g., Hoor et al., 2002; Pan et al., 2004). In the northern summer the ExTL extends up to 30 K in potential temperature above the local dynamic tropopause and between 20-25 K in all other seasons (Hoor et al., 2004). In the southern hemisphere, the ExTL seems to be more shallow (Hegglin et al., 2009).

The vertical distribution of trace species in the extratropical UTLS crucially depends on the large scale stratospheric circulation (e.g., Butchart, 2014) and on stratosphere-troposphere exchange (STE) across the tropopause (e.g., Holton et al., 1995; Stohl et al., 2003; Sprenger et al., 2003). Overall, there are three prominent pathways for air into the ExTL. First, the stratospheric circulation affects the ExTL through the descent of old stratospheric ozone rich air in the deep branch into the UTLS. Second, relatively young air from the tropical and subtropical UTLS can enter the region of the extratropical UTLS through fast

two-way eddy mixing in the shallow branch of the stratospheric circulation. A recent study based on airborne measurements showed the effect of these two competing transport pathways on the abundance of carbon monoxide (CO) over the course of the Arctic winter in the ExTL (Krause et al., 2018). A third pathway into the ExTL is direct injection of extratropical tropospheric air into the stratosphere by troposphere-to-stratosphere transport (TST). In the extratropics STE predominantly occurs in association with processes in baroclinic planetary- and synoptic-scale waves (Sprenger et al., 2003, 2007), small scale gravity

waves (Langford et al., 1996; Whiteway et al., 2003; Miyazaki et al., 2010), and in regions affected by convective systems (e.g., Poulida et al., 1996; Tang et al., 2011; Homeyer, 2015). Together these processes shape the vertical profiles of the trace species and thus have an impact on the surface climate trough a radiative feedback mechanism. Changes in the vertical abundance of radiatively active trace species in the UTLS, for instance for water vapor and ozone, relatively have the largest impact on the surface temperature (Riese et al., 2012).

Climatological studies revealed that in the northern hemisphere mid latitudes STE occurs predominantly in regions of enhanced cyclone activity, the storm tracks, over the North Atlantic and Pacific as well as over the Mediterranean Sea (Sprenger et al., 2003). Generally, in the extratropics STE occurs more frequently during winter and more mass is transported from the stratosphere into the troposphere, i.e., stratosphere-to-troposphere transport (STT) than vice versa, i.e., TST. This has recently

been reported by Škerlak et al. (2014) who analyzed STE based on a 33 year long time period of ERA-Interim reanalysis data and trajectory calculations. This analysis further confirmed earlier studies with respect to spatial and temporal occurrence of STE (e.g., Chen, 1995; Morgenstern and Carver, 1999; Dethof et al., 2000). These results are independent of the definition of the tropopause as shown by Boothe and Homeyer (2017) who used four different modern reanalysis data sets to analyze STE



as well as lapse-rate and dynamic tropopause definitions.

In the extratropics Rossby waves are crucial for STE. During their life cycle, Rossby waves can lead to the formation of so called stratospheric streamers and cut-off lows, both being regions of strong STE activity (Sprenger et al., 2007). STE occurs
also in tropopause folds along jet streams due to the cross-frontal secondary circulation. Sprenger et al. (2003) demonstrated that exchange between subtropical and extratropical air masses across the subtropical jet predominantly occurs in these folds, but tropopause folds exists also at higher latitudes (Škerlak et al., 2015). Idealized simulations of baroclinic life cycles and analyses of reanalysis data showed that dynamic instabilities with low Richardson numbers, thus large vertical shear of the horizontal wind, lead to mixing around tropopause folds (Bush and Peltier, 1994; Jaeger and Sprenger, 2007). Using similar
data and methods as Škerlak et al. (2014), Reutter et al. (2015) studied the relevance of extratropical cyclones for STE over the North Atlantic. They found that the STT mass flux is generally larger than the TST mass flux and that the region of the exchange varies slightly during the life cycle of the cyclone but most exchange takes place around the cyclone center in regions with low tropopause. This study confirmed earlier findings which suggested that STE occurs close to the cyclone center or rather in regions with relatively low tropopause, thus more on the cyclonic side of the jet in the region of the trough (e.g.,
Wernli and Davies, 1997).

All processes which lead to cross tropopause transport of air parcels have one common impact on this air parcel, they change its potential vorticity (PV). Such PV non-conserving processes must act on the air parcel and are related to radiation, phase changes of water in the atmosphere, and friction (Hoskins et al., 1985). Lamarque and Hess (1994) separated between diabatic,
i.e., potential temperature changing, and diffusive, i.e., related to friction, processes and showed that diabatic processes play a more vital role for STE than diffusive processes. Radiative effects may play an important role in anti-cyclones (Zierl and Wirth, 1997) where radiation lowers the tropopause and thus leads to a mass flux from the troposphere into the stratosphere. Radiation is also a key process to dissolve stratospheric cut-off lows in the troposphere (Forster and Wirth, 2000). Clouds and related diabatic heating may also have an impact on STE. For instance warm conveyor belts, i.e. airstreams ahead of cold
fronts associated with extratropical cyclones in which strong diabatic heating by latent heat release occurs (e.g., Wernli and Davies, 1997), can reach the upper troposphere and modify the PV, consequently allowing for exchange between tropospheric and stratospheric air (Wirth, 1995; Wernli and Bourqui, 2002). Similar, rapid transfer from the boundary layer into the UTLS is evident in convective systems, which sometimes have the potential to overshoot into the stratosphere (Poulida et al., 1996; Stenchikov et al., 1996; Homeyer et al., 2014). Convection can also trigger gravity waves which occasionally break or dissipate
and thus lead to small scale mixing between tropospheric and stratospheric air masses (Whiteway et al., 2003). Jet-induced gravity waves might play a vital role for STE on small scales, inducing turbulence and consequently allow for mixing between adjacent atmospheric layers (e.g., Langford et al., 1996; Lamarque et al., 1996). Strong shear zones are often apparent at the edges of the jet-streams which lead to filamentation of tropospheric or stratospheric air masses (Appenzeller et al., 1996). In these shear zones Kelvin-Helmholtz instabilities can emerge and lead to intense turbulence (e.g., Pepler et al., 1998; Whiteway
et al., 2004). In turn this can lead to mixing of air masses around the jet streams. However, this mixing is thought to be more





relevant at the lower edge (e.g., Danielsen, 1968; Shapiro, 1980) and on the cyclonic side of the jet, thus rather in the trough than in the ridge of baroclinic waves (e.g., Pan et al., 2007; Konopka and Pan, 2012).

Only a few studies focused on mixing and STE on the anti-cyclonic side of jet in the ridges of baroclinic waves. Early suggestion were based on individual airborne observations with small scale waves being responsible for cross tropopause transport in these regions (Shapiro, 1980; Danielsen et al., 1991). Model simulations by Lamarque and Hess (1994) showed that PV is not conserved in the ridge of the studied baroclinic wave and that cloud related processes lead to STE. Forster and Wirth (2000) showed how radiative effects can affect the tropopause altitude in anti-cyclones and thus lead to mass exchange from the troposphere into the stratosphere over the course of several days. Recently, Kunkel et al. (2016) showed that turbulent motions can occur on top of warm conveyor belt outflows at the altitude of the tropopause. This coincidence of turbulence in a region of the so called tropopause inversion layer (TIL. Birner et al., 2002) addresses an open question in the extratropical UTLS which is in particular relevant for the ExTL: Does the TIL affect the formation of the ExTL by inhibiting mixing or STE?

This study picks up the idea of turbulent mixing in regions of enhanced lower stratospheric static stability which initial resulted from experiments of baroclinic life cycles (Kunkel et al., 2016) and which has recently been described by Kaluza et al. (2018) in composites of baroclinic waves over the North Atlantic. Birner et al. (2002) and Grise et al. (2010) both discussed that enhanced values of static stability and shear zones emerge in close vicinity at the tropopause level. Zhang et al. (2015) addressed this further by linking the enhanced wind shear to propagating inertia-gravity waves with the potential to induce mixing, however, without addressing the larger scale meteorological conditions explicitly.

In this study we aim to analyze whether those turbulent signatures lead to mixing and potential exchange of tropospheric and stratospheric air masses in the ridge of baroclinic waves. For this we use complementary data of airborne measurements, numerical weather forecast data, trajectory calculations, and idealized baroclinic life cycle experiments which are introduced in Section 2. We will first focus on a research flight from the Wave-Driven Isentropic Exchange campaign (Section 3) which aimed to measure chemical constituents and state parameters across the tropopause in a baroclinic wave over the North Atlantic. We then extend a set of well known idealized simulations of baroclinic life cycles to analyze STE and to obtain a comprehensive understanding of the processes which lead to mixing and potential STE in the ridge of the baroclinic waves (Section 4). We finalize our study with a summary and a conclusion in Section 5.

## 2  Data and methods

### 2.1  Measurements during the WISE campaign 2017

In autumn 2017 the airborne research mission Wave-driven ISentropic Exchange (WISE) took place from Oberpfaffenhofen, Germany and Shannon, Ireland with the German HALO (High Altitude Long range) research aircraft. The main goals of the mission are to examine mixing processes in the UTLS in association with Rossby wave breaking and to study the impact of the



Asian Summer Monsoon circulation on the budget of radiatively active species in the lower stratosphere. One specific target is to study the relation between the lower stratospheric static stability and cross-tropopause exchange in the extratropics.

HALO was equipped with a unique set of instruments for in-situ and remote sensing measurements. In this study we use in-situ measurements of CO and $N_2O$, and potential temperature $\Theta$. CO and $N_2O$ have been measured with the University of Mainz Airborne Quantum Cascade Laser Spectrometer (UMAQS). The instrument is based on direct absorption spectroscopy using a continuous-wave quantum cascade laser with a sweep rate of 2 kHz (Müller et al., 2015). For the WISE campaign the total drift-corrected uncertainty were determined to be $0.94\,\mathrm{ppb_v}$ for CO and $0.18\,\mathrm{ppb_v}$ for $N_2O$. Basic state parameters such as temperature, pressure, the three-dimensional wind vector and others are measured with Basic HALO Measurement and Sensor System (BAHAMAS). The system is part of the basic aircraft and consists of a data acquisition system and a suite of sensors for basic meteorological and aerodynamic measurements. The system also contains interfaces into several aircraft systems like the inertial reference unit or the air data computer in order to monitor aircraft state parameters (Krautstrunk and Giez, 2012). The nose boom of HALO is part of the system and carries the air data probe for pressure and air flow measurements which are needed for determination of the wind vector. Additional BAHAMAS installations are six Total Air Temperature (TAT) housings on the aircraft nose which can be used for temperature measurements and as inlets for sensors inside the nose. Two of these housings contain an open wire PT100 resistance thermometer for atmospheric temperature measurements thus providing redundancy for this important parameter. The basic frequency for all atmospheric units is 100 Hz, data is usually processed on a 10 Hz basis. The accuracy of the pressure measurement is 0.3 hPa while the accuracy of the static temperature measurement is 0.5 K. We also will show remote sensing measurements from the Gimballed Limb Observer for Radiance Imaging of the Atmosphere (GLORIA), which provides profile information of temperature and static stability, in addition to numerous trace gases. GLORIA is an airborne infrared limb imager combining a two-dimensional infrared detector with a Fourier transform spectrometer (Friedl-Vallon et al., 2014; Riese et al., 2014). The viewing direction is to the right of flight direction and depending on flight altitude and flight direction GLORIA observes the atmosphere between about 5 km and flight altitude with a vertical sampling of about 150 m at a tangent altitude of 10 km. The vertical resolution of retrieved temperature profiles and static stability is of the order of 300 m. The horizontal sampling along the flight track is up to 2 km (Kaufmann et al., 2015; Ungermann et al., 2015).

In total the WISE campaign comprised more than 140 flight hours during 15 research flights between 12.09.2017 and 21.10.2017. All flights except the first two started in Shannon, Ireland and covered the North Atlantic between Greenland, Newfoundland, the Azores, and Europe as well as continental Western and Northern Europe. HALO was mostly flying in the UTLS up to ceiling altitudes of about 15 km, which corresponds to maximum potential temperature values of about 405 K and minimum pressure values of 130 hPa. Research flight 07 (RF07) on 28.09.2017 had the goal to study the abundance of trace species in the extratropical tropopause region in relation to the occurrence of enhanced values of static stability in the lower stratosphere and to potential STE. Such conditions in the ridge of extratropical baroclinic waves resemble the predictions of the idealized simulations from Kunkel et al. (2016). Thus, the flight was planned in the ridge of a synoptic-scale baroclinic wave




which evolved during the previous days over the North Atlantic at the edge of a larger scale trough. A detailed description of the synoptic situation and the flight path will be given in Sec. 3.1.

## 2.2 ECMWF forecast data

We use forecast data from the European Centre for Medium-Range Weather Forecast (ECMWF) to support the analysis of the airborne measurements and to provide more background information about the synoptic situation. We choose to use forecast data from ECMWF because it is hourly available at a very fine horizontal resolution. The forecast starts at 28.09.2017 00 UTC and we use the first 36 hourly steps until 29.09.2017 12 UTC. This data is used for analysis along the flight path where the full resolution is used corresponding to regular longitude/latitude grid spacing of about $0.07°$. Moreover, we use a slightly degraded data set with a horizontal grid spacing of $0.125°$ in the horizontal for a trajectory analysis. The forecast data has 137 vertical hybrid pressure-sigma levels up to 0.01 hPa with a vertical spacing of roughly 300 m in the UTLS.

## 2.3 Idealized baroclinic life cycle experiments

We complement our analysis of the airborne measurements using results from idealized baroclinic life cycle experiments. We continue the work of Kunkel et al. (2016) using the same setup of the COSMO model (Steppeler et al., 2003) but extend the analysis with a more specific focus to analyze STE. For this we included additional artificial tracers to mark air masses which are initially located either in the troposphere or in the stratosphere, separated by the dynamic tropopause for which we use the isosurface of 2 pvu ($1\,\mathrm{pvu} = 10^{-6}\,\mathrm{K\,m^2\,kg^{-1}\,s^{-1}}$). Moreover, we included a tracer which is passively advected and which carries the information of the initial value of potential vorticity. With this tracer it is possible to determine how much PV in each model box has changed by diabatic processes since model start (Kunkel et al., 2014). Evaluating the difference between the current PV and the advected initial PV at the dynamic tropopause allows then to detect regions of TST and STT.

We then repeated simulations from Kunkel et al. (2016) which included parameterizations for large scale and convective clouds, radiation and turbulence which have been labeled with BRTC (BRTC = Bulk microphysics, Radiation, Turbulence, and Convection) by these authors. The model grid has a regular horizontal grid spacing of $0.4°$ in longitude and latitude and a vertical grid spacing of 110 m in the UTLS. Since the meteorological situation during RF07 was dominated by wave breaking event strongly resembling a life cycle 1 (LC1), we will focus our discussion in Section 4 to results from LC1 experiments (Thorncroft et al., 1993), but we note that we also conducted simulations of life cycle 2 (LC2). Moreover, the LC2 experiments qualitatively gave the same results as the LC1 experiments. More information about the model setup and physics is given in Kunkel et al. (2016) and references therein.

## 2.4 Trajectory calculations with LAGRANTO

The LAGrangian ANalysis TOol LAGRANTO (Sprenger and Wernli, 2015) allows calculating trajectories using the kinematic wind from four dimensional meteorological data. It is possible to use both ECMWF and COSMO data as input for the tra-





jectory calculations. The first trajectory analysis is based on ECMWF forecast data which provides wind fields every hour. Trajectories have been initialized along the flight path with the goal to obtain a more comprehensive picture of the temporal evolution of meteorological parameters of the measured air masses. We will give more details on the trajectory start points and the analysis in Section 3.3.

For the second trajectory analysis we use COSMO wind fields. COSMO output is available every hour on a horizontal grid with spacing of $0.4°$ and a vertical spacing of about 110 m in the UTLS, thus providing a high resolution input grid for the trajectory calculations. Based on the COSMO model output the goal of the trajectory calculations is to identify regions of cross tropopause transport in the baroclinic life cycle experiments and whether exchange trajectories are evident in the ridge of the

baroclinic wave. More details will be given in Section 4.

In general, we identify the exchange of air masses between the stratosphere and troposphere by the evolution of the PV along a trajectory. In this study we use the 2 pvu isosurface as dynamic tropopause. If the PV of an air parcel increases from values below 2 pvu to values above this threshold, we mark this trajectory as a TST-trajectory and vice versa for STT.

## 3 Quasi-isentropic mixing in a ridge of a baroclinic wave during WISE

### 3.1 Synoptic situation and flight plan of WISE RF07

WISE RF07 targeted a fast evolving baroclinic wave which emerged at the southern tip of a large scale trough in the central North Atlantic in the early hours of 27.09.2017. While the large scale trough traveled relatively slow over the North Atlantic,

the small scale wave evolved rather fast, with a formation of a warm sector and well defined upper tropospheric fronts within 24 hours as indicated by the PV at 330 K at 15:00 UTC on 28.09.2017 (Figure 1a).

The flight focused on the rapidly evolving ridge, which was expected to be strongly affected by diabatic processes and vertical transport of boundary layer air in region of the warm conveyor belt (WCB) ahead the surface cold front. This system was

expected to be situated close to Ireland in the afternoon hours of 28.09.2017. Take-off of WISE RF07 was at 13.15 UTC with a total flight time of 7h 47min. The flight strategy was twofold: first stagged flight levels at FL400 (north-eastward), FL380 (south-westward), FL360 (north-eastward), FL340 (south-westward), FL420 (north-eastward), and FL450 (south-westward) through the ridge of the baroclinic wave and second a survey of the stratospheric air with large PV values above the occlusion of the baroclinic wave west of Ireland (flight path in Figures 1a,b and 2a). The intention of the first part in the northernmost part

of the ridge of the baroclinic wave was to conduct co-located measurements of atmospheric state parameters and trace species across the local tropopause. The stagged flight level allowed to measure these quantities in-situ at different altitudes below, at, and above the tropopause. From the highest flight levels (FL420 and FL450) in cloud-free conditions two dimensional distributions of temperature and trace species along the flight track were remotely measured with GLORIA. Our analysis focuses





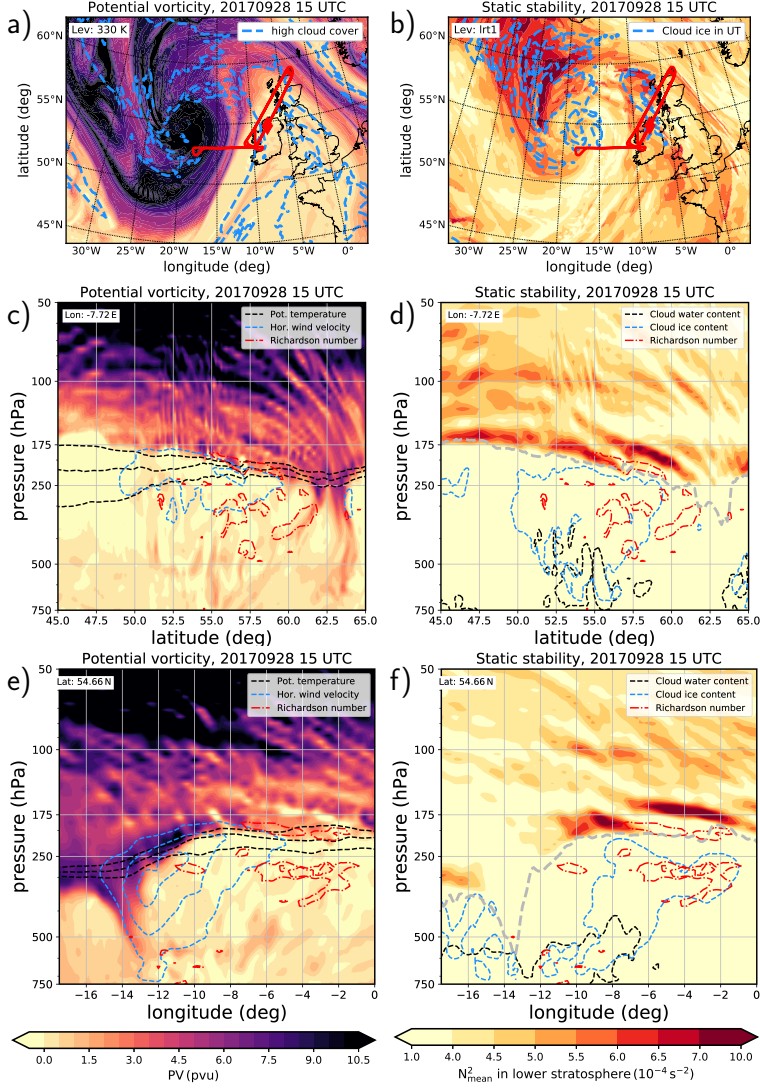

**Figure 1.** Synoptic situation during WISE RF7 over the Western North Atlantic at 15 UTC on 28.09.2017. Left panels show potential vorticity a) on the isentropic surface with 330 K with high cloud cover (blue dashed marks a value of 0.9), c) along $-7.72\,°$E longitude, and e) along $54.66\,°$N latitude. Panels c) and e) include isolines of potential temperature (black, values of 330 K, 335 K, 340 K), of horizontal wind velocity (blue, values of 35 ms$^{-1}$, 45 ms$^{-1}$, 55 ms$^{-1}$, and 65 ms$^{-1}$), and of the Richardson number (red, value of 1.0). Right panels show static stability (in $10^{-4}\,\mathrm{s}^{-2}$) b) as the mean value between the local thermal tropopause and 1 km above with integrated cloud ice column between 1.5 km below the local thermal tropopause, i.e., the first lapse-rate tropopause (lrt1) and this level (blue dashed, value of $2.5 \times 10^{-3}\,\mathrm{kg\,m}^{-2}$). Panels d) and f) show cross sections of static stability accordingly to panels c) and e) with isolines showing cloud water content (black, values of $5 \times 10^{-6}\,\mathrm{kg\,kg}^{-1}$), cloud ice content (blue, values of $5 \times 10^{-6}\,\mathrm{kg\,kg}^{-1}$), and the Richardson number (red, value of 1.0). The first lapse-rate tropopause is defined following WMO (1957).




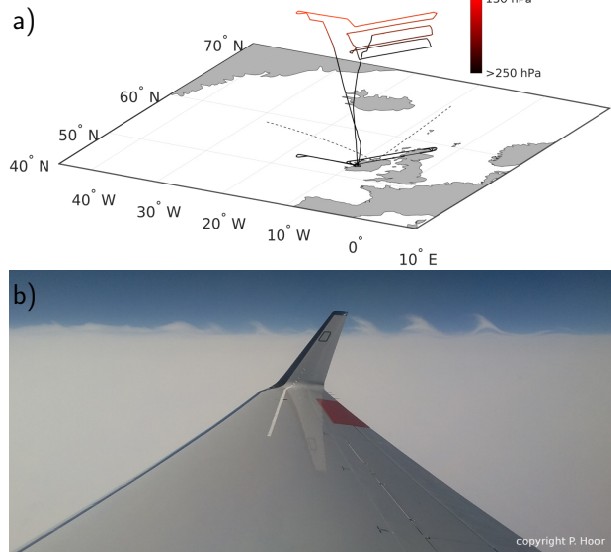

**Figure 2.** a) Three dimensional flight track of WISE RF07, color-coded with pressure (in hPa) and the angle of view of the photo shown in b) indicated by the dashed lines on the surface. b) Photo taken on board HALO at 15:03 UTC during WISE RF07 showing Kelvin-Helmholtz cloud billows on top of a cirrus cloud deck.

on the first part of the RF07, in particular on the first flight leg in south-westward direction at FL380.

A large part of the first leg at FL380 in south-westerly direction was close above a wide-spread cirrus cloud deck associated with the upper tropospheric outflow of the WCB of the low pressure system. The high cloud cover in Figure 1a shows the location of this cloud deck. The WCB manifests also in the low values of PV in the upper troposphere (Figures 1c,d). PV values are close to 0 pvu or even slightly below which is common in such a situation due to a decreasing heating rate above the level of maximum heating (e.g., Chagnon et al., 2013; Joos and Wernli, 2012).

The PV shows enhanced variability above the WCB outflow region in the lower stratosphere (Figures 1c,e). These structures are even more evident in static stability (Figures 1d,f). Maximum values are above $10 \times 10^{-4}\,\mathrm{s}^{-2}$ with almost tropospheric background values in between. Kunkel et al. (2016) linked these large values of static stability to moist tropospheric dynamics related to the WCB ascent and to the radiative feedback from cirrus clouds close to the (thermal) tropopause. Additional contributions might result from propagating inertia-gravity wave in the lowermost stratosphere (Kunkel et al., 2014). In parts of the regions of enhanced static stability in the lowermost stratosphere low Richardson numbers emerge with values below 1. These low values suggest the potential occurrence of turbulent motions in the region of the tropopause. Such a co-located enhancement of static stability and presence of turbulence was also evident in the life cycle experiments of Kunkel et al. (2016) and




has recently been reported by Kaluza et al. (2018) based on a composite analysis of baroclinic waves over the North Atlantic.

Another indication of turbulence in the region above the cloud deck stems directly from observations of Kelvin-Helmholtz cloud billows close to the flight path (Figure 2b). A photo was taken at 15.03 UTC in a north-western direction (Figure 2a, diamond in Figures 1a,b). The billows are indicative for a Kelvin-Helmholtz instability (KHI) which is thought to favor mixing of adjacent atmospheric layers and thus affects the vertical gradient of trace species. Notably, since the KHI emerges in the vicinity of the tropopause, it could potentially lead to STE. Commonly, a critical Richardson number of 0.25 identifies a KHI. However, in a non-convective situation such small Richardson numbers rely on large vertical shear of the horizontal wind. In turn, models based on a discretized grid with grid spacings of a few hundred meters or more in the vertical can often not resolve the vertical shear sufficiently. In the region of the observed cloud billows we find Richardson numbers in the ECMWF model of about 1. We therefore use a Richardson number equal 1 as a proxy for KHI in the UTLS for our analysis.

Now we focus on the flight leg from north-east to south-west at FL380 between 14.20 and 15.20 UTC. HALO came from FL400 which was then deeper in the stratosphere with potential temperatures above 350 K. At FL 380 HALO was flying in the lowermost stratosphere, gradually approaching and eventually crossing the dynamic tropopause horizontally slightly above the dynamic tropopause which was slightly tilted according to the ECMWF analysis (Figure 3a). This is in remarkable agreement with measurements of $N_2O$, which increases to tropospheric values on FL 380 (Figure 4) as discussed further below. On this leg the aircraft crossed the lower part of the structures with alternating large and low values of static stability. This structure is potentially linked to a propagating inertia-gravity wave which commonly occur during baroclinic wave developments (e.g., O'Sullivan and Dunkerton, 1995). Furthermore, in the region with low values of static stability an area with low Richardson numbers is evident which extends into the region of the maximum of static stability around 15 UTC.

Before we analyze the time period between 14:00 and 15:00 UTC more specifically in Section 3.2, we want to point out one model deficiency. Although the ECMWF forecast predicted the atmospheric state very well, there is evidence that extreme values of state parameters are missed or at least underestimated with potential consequences on the representation of mixing in the UTLS. We already mentioned that we think that the Richardson numbers from the ECMWF forecast might have too large values in some regions around the tropopause. However, this is difficult to verify from airborne measurements, since the full two dimensional information on the temperature and wind are missing along the flight path. However, using temperature retrievals from GLORIA, we can at least compare static stability (the numerator of the Richardson number calculation) from the model and from measurements (Figure 3b,c). For this we use the last north-eastward leg on FL420, i.e., $\sim$ 13 km and 169 hPa, between 17:20 and 18:20 UTC, when HALO was flying above the maximum values of static stability. Static stability from GLORIA measurements exhibits larger values than from the ECMWF model as well as a smaller vertical extension of the entire wave structure. Thus, the model forecast underestimates the strength of the inversion, most potentially due to defincies in representing the gravity wave in this region. Since the gravity wave also affects the three dimensional wind, it could be assumed that the vertical shear of the horizontal wind might not be sufficiently represented in the model and thus the Gradient





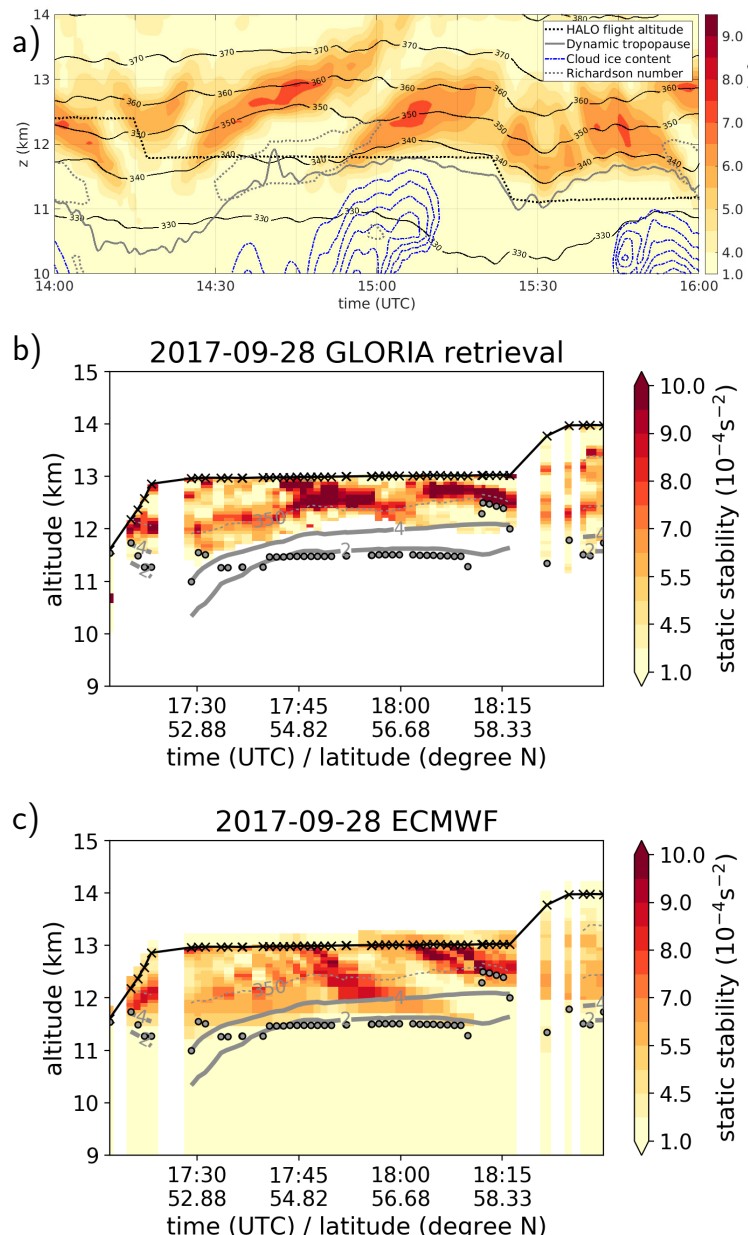

**Figure 3.** a) Static stability along the flight path between 14:00-16:00 UTC (color-shaded) along with potential temperature (black solid), cloud ice water content (blue dashed), the altitude of the dynamic tropopause (gray solid) based on ECMWF forecast data. Black dotted line shows the altitude of HALO. b) Static stability at GLORIA tangent points between 17:20-18:20 UTC. Black line shows the flight altitude with black crosses marking points of measurement. Thick gray line show 2 and 4 pvu, thin gray line the 350 K isentrope based on ECMWF forecast data. Grey dotted points mark the location of the lapse rate tropopause. c) ECMWF forecast data sampled at GLORIA tangent points.



Richardson number.

In summary, the synoptic situation during WISE RF07 strongly resembles the situation of the idealized baroclinic wave of Kunkel et al. (2016). Consequently, it is now possible to investigate a prediction from an idealized model study with airborne
measurements and seek for signs of turbulent motions in the measurement data.

### 3.2    Airborne in-situ measurements and evidence of mixing around the tropopause - WISE RF07

We start our analysis by focusing on the time period between 14:20-14:54 UTC (gray areas in Figure 4) and will first concentrate on the in-situ measurements of nitrous oxide $N_2O$ and potential temperature $\Theta$ along with several parameters from the ECMWF forecast interpolated on the flight track. Based on $N_2O$ and $\Theta$ we further subdivided the time period of interest into three
shorter periods (gray scales in Figure 4). During the first period from 14:20-14:46 UTC $N_2O$ volume mixing ratios below $331.31 \pm 0.45\,\mathrm{ppb_v}$ indicate that HALO was flying in an airmass of stratospheric origin. This value represents the airborne measured tropospheric mean of $N_2O$ during the WISE campaign and is calculated following Müller et al. (2015). HALO was flying at a constant pressure level while slowly approaching the dynamic tropopause. In the second time period between 14:36-14:48 UTC the distance between tropopause and flight level decreased to a few hundred meters or less according to the PV
analysis. Notably, this part of the flight close to the tropopause in the ExTL shows low Richardson numbers. Around 14:40 UTC the time series of both $N_2O$ and $\Theta$ show strong wave-like structures with periods of about 2 minutes. The horizontal wavelength can be roughly estimated to be about 25 km, using a mean ground speed of 210 $\mathrm{ms^{-1}}$ of HALO during this part of the flight and assuming that the wave has been crossed perpendicular to the wave crests. The amplitude of the wave is about 2.5 K and the wave spans the potential temperature range between 335-340 K. Interestingly, this wave structure is hardly
evident in the modeled $\Theta_M$, indicating that the model might have issues representing these scales accurately. The model shows signs of gravity wave activity, however, not directly at the flight altitude but slightly above (not shown). The ECMWF forecast model with a horizontal grid spacing of about 8 km can barely resolve this wave pattern with this fine scale structure due to the grid spacing of the model. The last period between 14:48-14:54 UTC shows a strong variability in both $N_2O$ and $\Theta$ without any clear wave signal but a large variability of $\Theta$ and $N_2O$ . This increase in variability could be regarded as a first indicator of
increased atmospheric turbulence.

If turbulence occurred in a region of strong tracer gradients at or upwind of the measurements, it should have affected the composition of trace gases. Both $CO$ and $N_2O$ exhibit vertical gradients in the lower stratosphere (Figure 5a for $CO$). We first focus on $CO$ since it has a much stronger gradient at the tropopause due to its shorter life time compared to $N_2O$. HALO was
initially in the stratosphere with low values of $CO$ and at the end of the time period close to the troposphere with larger values of $CO$ (Figure 5b). The gradual transition between the more stratospheric to more tropospheric $CO$ values occurs at potential temperatures between 335-340 K. This $\Theta$ interval also shows low Richardson numbers in the ECMWF model (Figures 4, 3a). Moreover, the color code in Figure 5b reveals the consecutive alternation between low potential temperature/large $CO$ values and high potential temperature/low $CO$ values, indicating the impact of the small scale wave on the tracer and potential



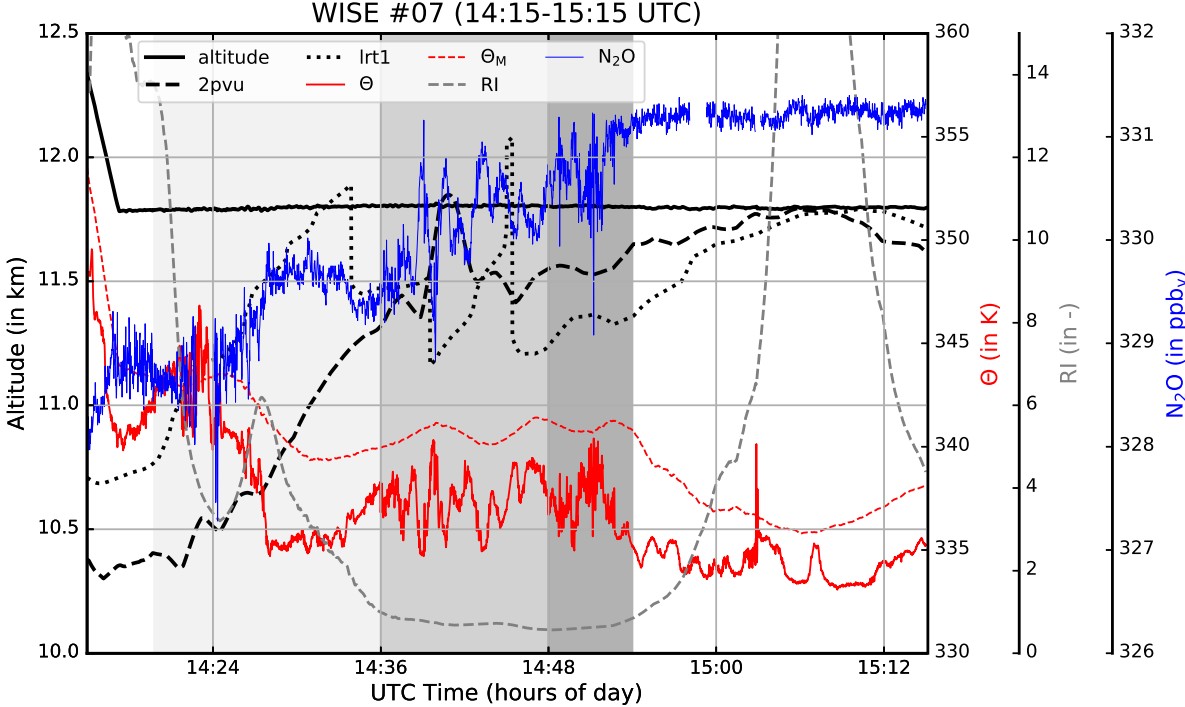

**Figure 4.** Time series of measured $N_2O$ (blue) and potential temperature $\Theta_M$ (red solid) as well as modeled potential temperature $\Theta$ (red dashed), Richardson number $RI$ (gray dashed), and altitude of the dynamic tropopause (black dashed) between 14:15 and 15:15 UTC during WISE RF07. Black solid line shows the altitude of the aircraft, the gray areas indicate the time periods discussed in the text.

temperature. To identify mixing of tracers, we analyzed the $N_2O$-CO relationship, which provides information on irreversible tracer exchange similar as $CO$-$O_3$ (e.g., Fischer et al., 2000). In general, at the tropopause the $CO - N_2O$ correlation starts with larger CO and larger $N_2O$ mixing ratios at potential temperatures typical for the extratropical tropopause. Above the tropopause with increasing potential temperature both $N_2O$ and CO mixing ratios decrease. However, the $CO - N_2O$ corre-

5   lation of the time period between 14:20-14:54 UTC does not show such a clear relationship of decreasing $N_2O$ and CO and simultaneous increasing potential temperature (Figure 5c). If the time period is instead divided into three shorter time periods following the gray shading in Figure 4, then only the first period shows a relationship between $N_2O$, CO and potential temperature as one would expect from the large scale vertical profiles of these quantities (Figure 5d,g). However, for the other two time periods the tracer-tracer relation with respect to potential temperature changes. For the time between 14:36-14:48 UTC

10   the vertical profile of CO shows a wave-like transition (Figure 5e), while the correlation shows isentropic mixing on different isentropes as indicated by mixing lines connecting tropospheric and stratospheric values between 335-340 K (Figure 5h). In the last time period the relation between $N_2O$, CO, and potential temperature seems to almost entirely break down, reflecting the initial thought of increased variability in the tracer mixing ratio and potential temperature (Figures 5f,i). Thus, based on the trace gas analysis, we think that turbulence increasingly affected the second and third time periods to a substantial degree in



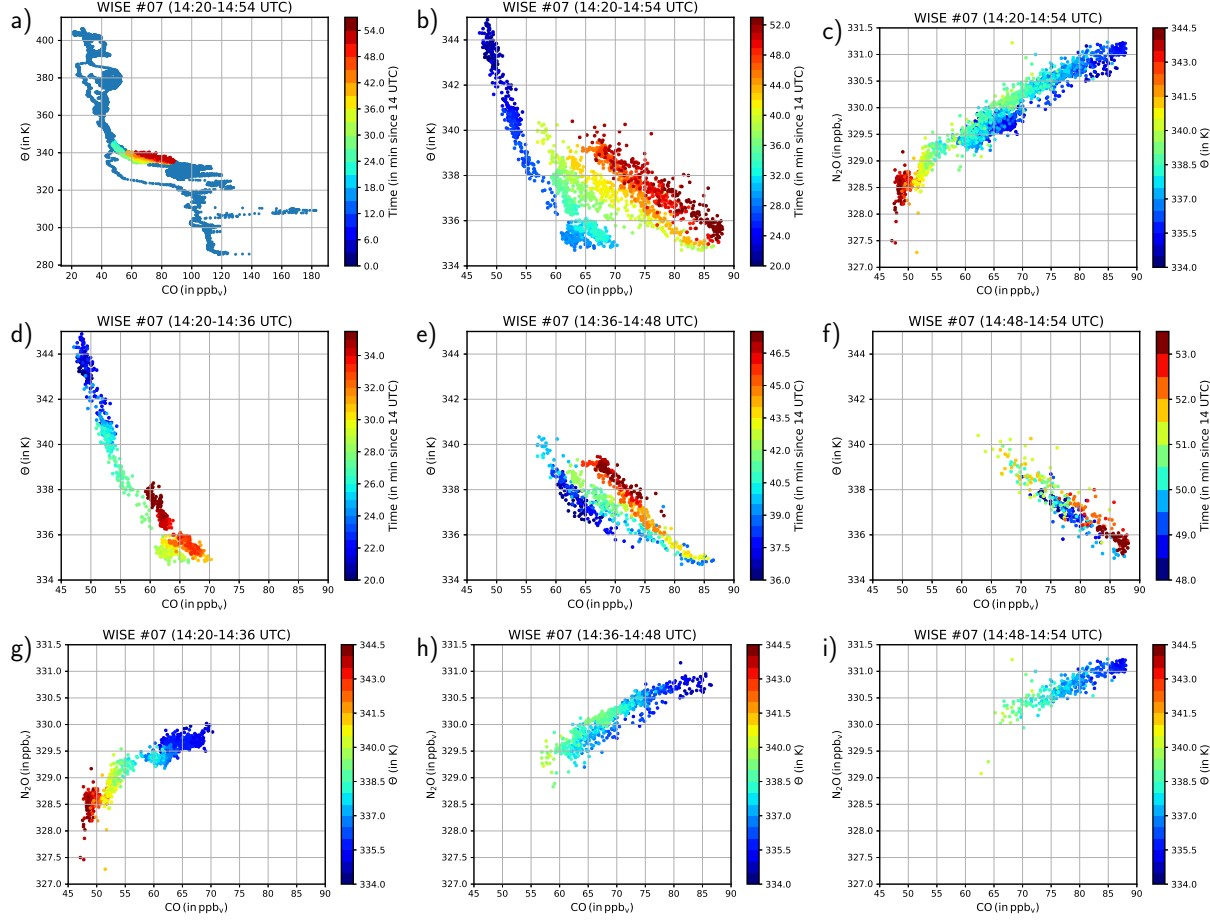

**Figure 5.** Trace gas analysis of the first south-westward leg on FL380, ie. 205 hPa of WISE RF07. a) Vertical profile of carbon monoxide for the entire flight (blue dots) and color-coded for the leg on FL380. b) Vertical profile of CO for the time period between 14:20-15:00 UTC, color-coded with time since 14:00 UTC. c) $CO - N_2O$ correlation for the time period 14:20-15:00 UTC, color-coded with potential temperature. d-f) Vertical profiles of CO for time periods between 14:20-14:36 UTC, 14:36-14:48 UTC, and 14:48-14:54 UTC, color-coded with time. g-i) $CO - N_2O$ correlations for the time periods between 14:20-14:36 UTC, 14:36-14:48 UTC, and 14:48-14:54 UTC, color-coded with potential temperature.



contrast to the first time period. This could ultimately lead to exchange of trace constituents across the tropopause in a narrow range of potential temperature levels.

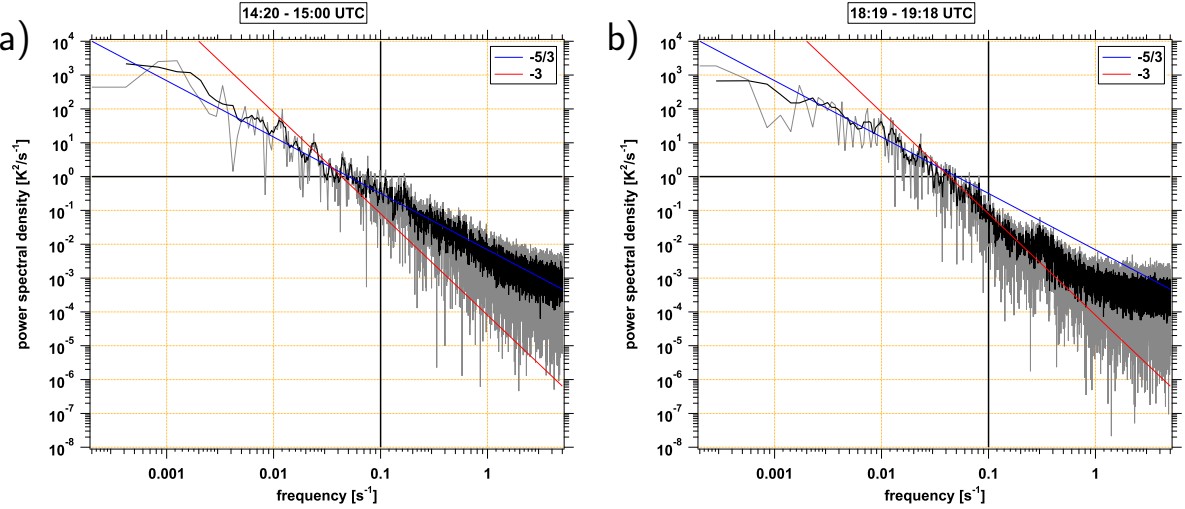

**Figure 6.** Power spectral densities of measured potential temperature from HALO for selected time periods: a) 14:20 - 15:00 UTC and b) 18:19 - 19:18 UTC. Black solid lines are introduced for better comparability, red solid lines shows a line with a slope of $k = -3$, blue solid line shows a line with a slop of $k = -5/3$.

We further analyze the occurrence of turbulence with the help of power spectral densities (PSD) of potential temperature. Power spectral densities of trace species or state parameters allows to estimate how much energy is present at a particular spatial scale (Vallis, 2017). On board HALO the frequency of measurements is about 10 Hz for the state parameters such as temperature and wind, while it is about 2-3 Hz for $CO$ and $N_2O$, which allows to identify structures down to about 100 m. The shape of these power spectral densities allows to assess the contribution of individual scale ranges to the total energy spectrum and indicates the type of turbulence that affects the considered domain (Vallis, 2017). The power spectral density from the flight leg on FL380 (Figure 6a) reveals a slope close to $k = -5/3$. This slope often characterizes 3D isotropic turbulence, i.e., dynamic processes on or below the meso scale affect the flow (Tung and Orlando, 2003). At a later leg in the same direction of the flight on flight level FL420 between 17:20-18:20 UTC the power spectral density follows rather a slope of $k = -3$ (Figure 6b). This slope rather indicates a dominance of geostrophic turbulence and thus larger synoptic scales affecting the flow. Hence, this suggests that meso-scale processes, e.g., related to gravity waves, might be substantial to explain the dynamics in the tropopause region.

### 3.3 Trajectory based history of measured air masses during WISE RF07

The analyses of trace gas distributions, tracer-tracer correlations, and power spectral densities suggest that mixing is evident in the region between 335K and 340 K. In particular, the $N_2O$ mixing ratios reveal that tropospheric and stratospheric air masses





participate in mixing process. Our next goal is to elucidate the recent history of the measured air masses. For this we calculated kinematic trajectories back and forward in time which start each second along the flight path for the time period between 14:24 UTC to 14:54 UTC based on ECMWF forecast data available between 28.09.2017 00 UTC and 29.09.2017 12 UTC. More specifically, the trajectories start at the horizontal location of the airplane plus at adjacent locations $\pm 0.07°$ in longitudinal and

latitudinal directions to cover some of the uncertainty due to the gridded representation of the meteorological input variables and to increase the number of trajectories for statistical purposes. Moreover, trajectories start at each full isentropic level between 334 K and 341 K and not only at the flight altitude of HALO. This allows us to study the fate of the entire region which is subject to mixing according to the measurement.

Since we are interested in STE and mixing, we first search for those air masses which cross the tropopause around the time of the measurement and which encounter a KHI. For this we filter all trajectories which cross the dynamic tropopause and which have Richardson numbers smaller than 1 at some time. We note, however, that our STE criterion is rather weak, since we only require that the PV of the trajectories is below the PV threshold for the dynamic tropopause at the start of the analysis (28.09.2017 00 UTC) and above the threshold at the end (29.09.2017 12 UTC). Thus, we do not speak of TST and STT tra-

jectories in a classical meaning (Stohl et al., 2003). As mentioned above, we consider low Richardson numbers in the order of one as good proxies for KHI. We further study the three time periods of interest from the earlier analysis separately (Figure 4).

Interestingly, we find trajectories indicating upward transport which show the same behaviour in dynamic and thermodynamic quantities in each of the three time periods between 14:20 UTC and 14:54 UTC. These trajectories always follow a wave

like flow from the North Atlantic towards the British Islands before they turn anti-cyclonically towards central Europe (Figure 7). Minimum pressure is evident over the North Atlantic when the trajectories pass through the large scale trough. Over Ireland and the British Islands the trajectories rise again while encountering the region of the KHI with minimum Richardson numbers. Starting from this region the trajectories strongly decelerate in a region of alternating horizontal divergence. During this time the trajectories cross back and forth over the dynamic tropopause.

The motion back and forth across the chosen PV value for the dynamic tropopause becomes also evident from the PV along the trajectories (Figure 8a). The trajectories show no straight traverse from the troposphere into the stratosphere. However, several points shall be noted for the time close to the measurement briefly before 15 UTC. The static stability shows a maximum which is not evident in PV but accompanied by a strong decrease in relative vorticity towards anti-cyclonic flow (Figure 8b,e).

The maximum in static stability follows a period where the trajectories encounter alternating vertical wind which is a sign of flow through small scale waves. At the end of this time period signs of turbulent motions increase. Initially, the turbulence index increases due to vertical wind shear as well as stretching and shearing deformation (Ellrod et al., 1992). Later low values of the Richardson number emerge which are indicative for KHI. The appearance of turbulence is further associated with a slight increase in potential temperature which is on the order of 1-2 K, thus the process can be regarded as quasi-isentropic.






**Figure 7.** Trajectories crossing the 2 pvu isosurface and encountering a dynamic instability which cross the flight track between 14:36 – 14:48 UTC. Trajectories start on 28.09.2017 01:00 UTC and end on 29.09.2017 11:00 UTC. The four panels show the following quantities along the trajectories: a) pressure (in hPa), b) potential vorticity (in pvu), c) time since 15:00 UTC (in h), and d) Richardson number. In total each panel shows 12375 individual trajectories.





**Figure 8.** Time series of dynamic and thermodynamic quantities of mixing trajectories crossing the dynamic tropopause and experiencing a KHI. The time axis is relative to 28.09.2017 15 UTC. The eight panels show the following variables along the trajectories: a) potential vorticity (in pvu), b) static stability (in $10^{-4}\,\mathrm{s}^{-2}$), c) pressure (in hPa), d) potential temperature (in K), e) relative vorticity (in $10^{-5}\,\mathrm{s}^{-1}$), f) vertical wind (in $\mathrm{m\,s}^{-1}$), g) Richardson number, and h) turbulence index after Ellrod et al. (1992) (in $10^{-6}\,\mathrm{s}^{-1}$).





The major difference between the three time periods between 14:20-14:54 UTC is the number of occurrence of these characteristic mixing trajectories, relative to the dynamic tropopause. During the first part between 14:20 - 14:36 UTC, we find only about 1.7 % out of 69048 trajectories crossing the tropopause and have low Richardson numbers. This number increases to 23.9 % out of 51768 in the second part and to 21.9 % out of 25920 in the last part from 14:48 - 14:54 UTC. Notably, this

result does not depend on the choice of our PV value for the dynamic tropopause. This indicates that a rather thick layer around the tropopause up to 3.5 pvu is affected by this mixing process, in particular in the between 14:36 - 14:48 UTC according to the trajectory analysis. Moreover, there is also a tendency of air masses from above to be mixed downward. For this we select trajectories which initially have a PV value above and finally below the desired dynamic tropopause value. While being most evident in the second and third part, such trajectories are in general much less common than those with an increasing PV value

over the considered time period. In line with this is that the stratospheric influence on the measured air masses substantially decreases from the first to the third time period. While about two third of the trajectories originate in regions above 5 pvu in the first part, only about a quarter does in the second part and less than 0.5 % in the third part.

Consequently, the trajectory analysis shows that different air masses of tropospheric and stratospheric origin come together

in the second and third part of the considered time period. This is in line with the measured trace gas concentrations of $N_2O$ and CO. However, based on the trajectory analysis it is difficult to analyze whether STE and in particular TST in a classical meaning occurs that an air parcel crosses the dynamic tropopause from the troposphere into the stratosphere. We also performed longer trajectory calculations which, however, did not provide any further clear answer on whether STE in a classical meaning occurred or not. One reason could be that the model fails to correctly resolve the process with the consequence that a

clear STE can not be diagnosed from the trajectory calculations. In contrast, assuming that the model performed well, it could simply mean that in this specific case the mixing occurred only across the tropopause with no substantial STE taking place. Independently on which is the case, this process changes the gradients of the trace gases in this region which in turn is of importance for radiative transfer calculations (e.g., Riese et al., 2012).

**4   Mechanisms for mixing in ridges of baroclinic waves**

After the analysis of airborne measurements, a first questions arises how generic such a mixing process may be in ridges of baroclinic waves. We can answer this question at least partly when searching for this process in idealized baroclinic life cycle experiments which are generic counterparts of atmospheric baroclinic waves. If the process is evident in the experiments, it might occur frequently in the real atmosphere and thus be of significance. Furthermore, the idealized experiments allow us to

further analyze the physical processes which lead to mixing and potentially to STE. For this we use model results based on the idealized simulations already used in Kunkel et al. (2016). The simulations include non-conservative processes (in terms of PV) such as large scale and convective cloud microphysics, radiative effects from trace species and clouds, as well as vertical turbulence. These simulations were labeled with BRTC (BRTC = Bulk microphysics, Radiation, Turbulence, and Convection)





in Kunkel et al. (2016). We further conducted simulations for life cycles 1 and 2 (Thorncroft et al., 1993), however, we focus our discussion here on results for life cycle 1. We extended the BRTC simulations by including tracers to mark the air which was initially in the stratosphere or troposphere as well as to trace the initial PV distribution. Furthermore, we calculated kinematic trajectories to analyze STE.

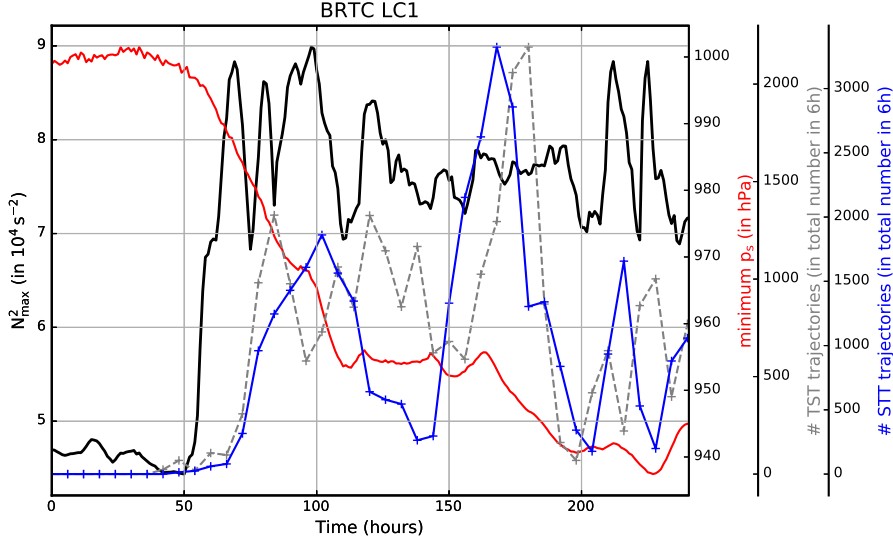

**Figure 9.** Temporal evolution of maximum static stability (black, in $10^{-4}\,\mathrm{s}^{-2}$), minimum surface pressure (red, in hPa) as well as the number of TST and STT trajectories per six hour interval over the course of the idealized baroclinic life cycle experiment BRTC LC1.

We first show that STE occurs in the model simulation. For this we calculated backward trajectories starting every six hours between 24 and 192 hours of model integration. The start points of these trajectories were distributed around the dynamic tropopause, i.e., the 2 pvu isosurface. They were initialized at each grid point in the horizontal and between 1.5 km below and 1.5 km above the local dynamic tropopause in the vertical. Each time more than 240.000 trajectories are then traced back

10 for six hours and consequently filtered based on whether they cross the tropopause. Furthermore, we searched for those STE trajectories which change their potential vorticity within the six hours by a given PV value to circumvent a residence time criterion suggested by Wernli and Bourqui (2002). For instance, we assume that trajectories which have an initial potential vorticity smaller than 1.5 pvu and a final one of larger 2.5 pvu have a larger propability to stay in the stratosphere than if the only criterion is to cross the 2 pvu isosurface.

The number of TST and STT trajectories based on these six hours long back trajectories reveal that STE starts to occur slightly after the time of the first enhancement of $N^2$ during the growing stage of the surface cyclone (Figure 9). Kunkel et al. (2016) attributed the increase of lower stratospheric static stability to updrafts in the troposphere. This can be regarded as the first time step when the tropopause is affected significantly by the tropospheric dynamics related to the baroclinic wave.





Note that the initial state of the baroclinic life cycle experiments is designed such that the tropospheric background value is $N^2 = 1 \times 10^{-4}\,\mathrm{s}^{-2}$ and that the stratospheric value is $N^2 = 4 \times 10^{-4}\,\mathrm{s}^{-2}$. Peak values for STE are evident after 160 h of model integration, in the final stage of the life cycle (Reutter et al., 2015). Thus, there is almost a temporal coincidence between the start of the enhancement of static stability and the first occurrence of STE.

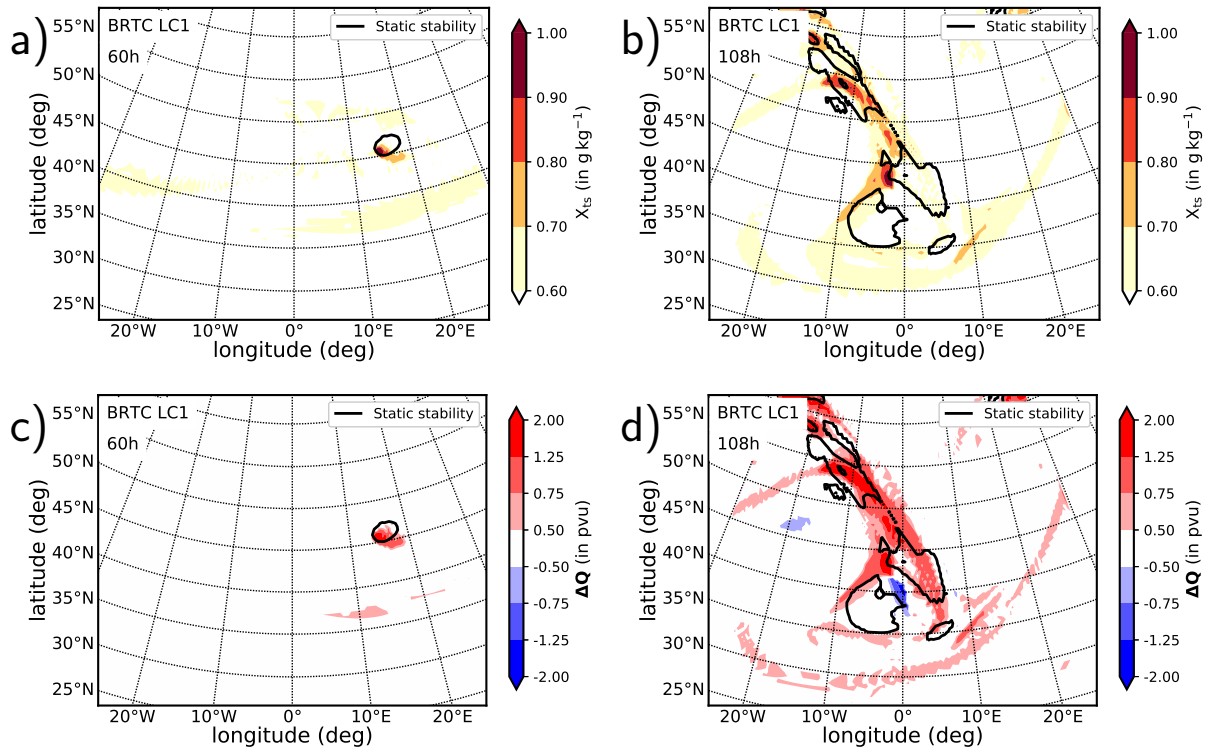

**Figure 10.** The tropospheric tracer $X_{\mathrm{ts}}$ on first full stratospheric level, i.e., the first full level with PV values larger 2 pvu for a) 60 h and b) 108 h after model start. Diabatic change of PV, $\Delta Q = Q - Q_{0,\mathrm{adv}}$, at the level of the dynamic tropopause with $Q_{0,\mathrm{adv}}$ being the advected initial PV and $Q$ being the full PV for c) 60 h and d) 108 h after model start. The black isoline shows static stability, $N^2 = 5.5 \times 10^{-4}\,\mathrm{s}^{-2}$.

We also find a spatial coincidence in the horizontal plane between the enhancement of $N^2$ above the thermal tropopause and TST across the dynamic tropopause by analyzing passive tracers in our idealized simulations (Figure 10). The tropospheric tracer, initialized with a constant non-zero mixing ratio in the troposphere and zero in the stratosphere, shows enhanced values at the first model layer in the stratosphere, exactly in the region where static stability is also enhanced. We define the first model

10  level in the stratosphere as the first full model level above the dynamic tropopause. This region also marks the region where the tropospheric tracer enters the stratosphere. This is first evident after about 60 h (Figure 10a), but also at later time steps (Figure 10b). We further can confirm these findings based on our diabatic PV tracer. This tracer carries the information about the difference between the current and the initial value of PV in each grid box (Kunkel et al., 2014). Evaluating this difference




on the dynamic tropopause allows us to diagnose whether an airmass at the dynamic tropopause gained or lost PV, thus whether this air mass initially resided in the troposphere or in the stratosphere. Positive values of this difference indicate a gain of PV, i.e., TST, and negative values a loss of PV, i.e., STT, of the respective air mass at the dynamic tropopause since model start (Figure 10c,d). In contrast to TST and although a temporal coincidence is also evident for $N^2$ enhancement and occurrence

5      of STT, no spatial co-occurrence is evident for STT in regions of enhanced $N^2$ (Figures 10c,d). Thus, from the distribution of these passive tracers, we can conclude that there is also a spatial coincidence between TST and enhancement of static stability.

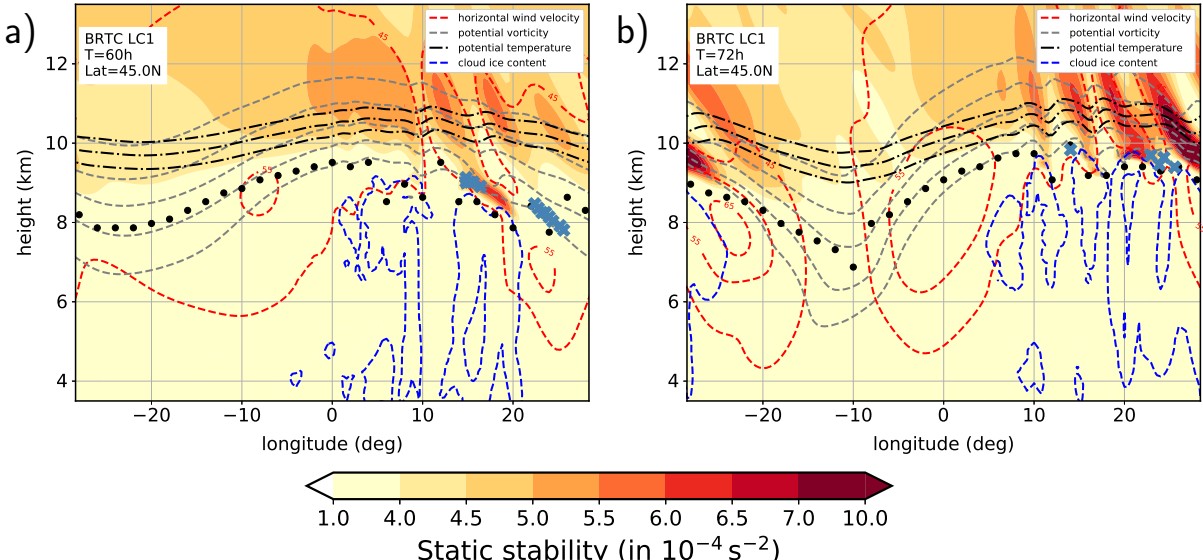

**Figure 11.** Zonal cross sections along the center of the model domain at 45 °N show static stability (contour filled), potential temperature (330 K, 335 K, 340 K, black lines), cloud ice water content ($10 \times 10^{-6}$ kg kg$^{-1}$, blue lines), potential vorticity (2-6 pvu, grey lines), as well as the altitude of the thermal tropopause (black dots). Red lines show isolines of the horizontal wind speed (45-65 ms$^{-1}$) and blue crosses show points of trajectories at the respective longitude which cross the tropopause from the troposphere to the stratosphere over the course of the last six hours for a) 60 h and b) 72 h after model start.

Furthermore, the exchange occurs in the ridge of the baroclinic wave, just above a region of ice cloud occurrence (Figure 11). This region is also strongly affected by a small scale wave pattern related to a propagating inertia gravity wave which

10     is evident in the isolines of potential temperature and PV. This wave is one source of the enhanced values of static stability. The other source is radiative cooling below the tropopause related to ice clouds in the upper troposphere. The first time TST occurs in this region is after about 60 h, while it is evidently more frequent during later stages of the life cycle. We thus found a pathway from the troposphere into the stratosphere in the ridge of baroclinic waves in our idealized simulations. The situation is as initially expected from the results of Kunkel et al. (2016) and resembles the situation of the mixing and potential TST in

15     the ridge during WISE RF07.



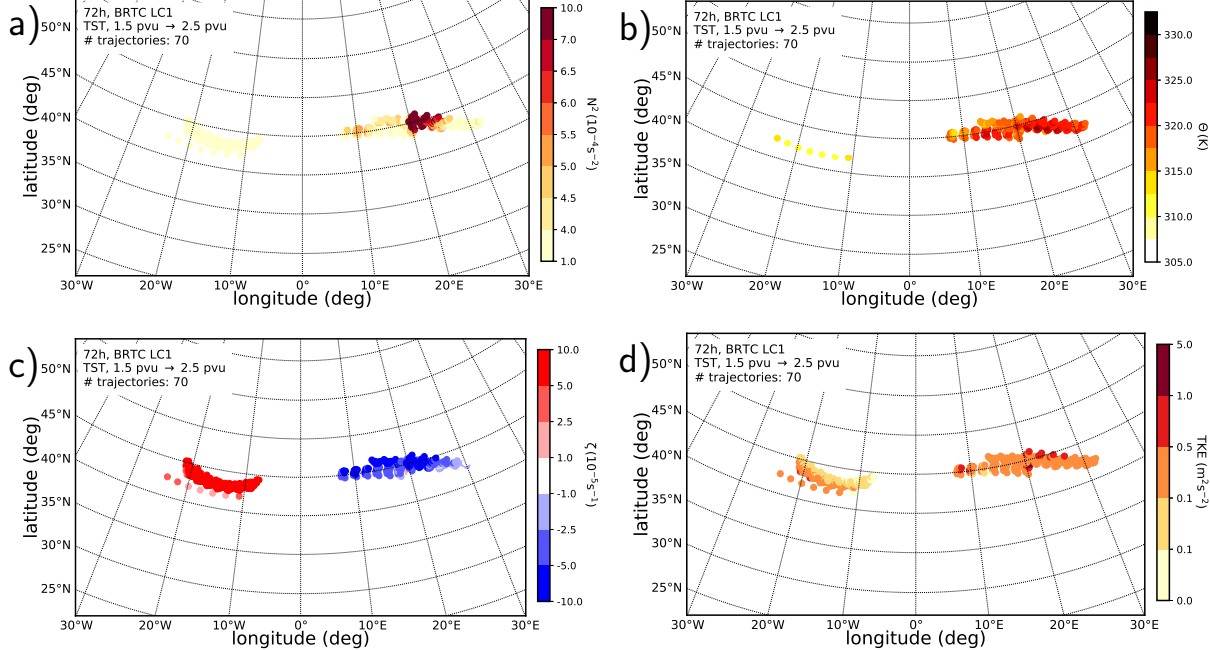

**Figure 12.** Trajectories starting 66 h and ending 72 h after model start which increase their PV from values smaller than 1.5 pvu to values larger 2.5 pvu within the six hour interval. Panels show a) static stability (in $10^{-4}\,\text{s}^{-2}$), b) potential temperature (in K), c) relative vorticity (in $10^{-5}\,\text{s}^{-1}$), and turbulent kinetic energy ($\text{m}^2\,\text{s}^{-2}$) along the trajectories for each hour.

Before we end our discussion by studying the processes leading to mixing, we want to highlight the fact that in our idealized simulations two type of TST trajectories are apparent (Figure 12). One set of TST trajectories exhibits low values of static stability, low potential temperatures, positive relative vorticity, and moderate turbulent kinetic energy (TKE). In contrast, the other set of TST trajectories shows large values of static stability, higher potential temperatures, negative relative vorticity, and larger values of TKE. Thus, while the first set of trajectories crosses the tropopause at the cyclonic side of the jet at rather low altitudes, the other set of trajectories crosses the tropopause at the anti-cyclonic side of the jet at rather high altitudes. Such trajectories are found over a large part of the life cycle, also in the case when we apply different criteria for a change of PV, e.g., from 1.5 pvu to 3.0 pvu. A common feature of the two sets of trajectories is that in both cases the potential temperature values hardly change in the six hours; thus, the TST occurs quasi-isentropically. We also note that the trajectories crossing the tropopause in the ridge of the wave experience large values of static stability, thus passoin through the tropopause inversion layer. Thus, what initially might seem counter-intuitive, i.e., the exchange in the vicinity of large values of static stability inhibiting vertical motions, is nevertheless possible in the ridge of the baroclinic waves.

The last point that we want to address are the processes causing the mixing and exchange across the tropopause in the ridge of the baroclinic wave. The occurrence of turbulence becomes apparent by low values of the Gradient Richardson number and

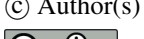



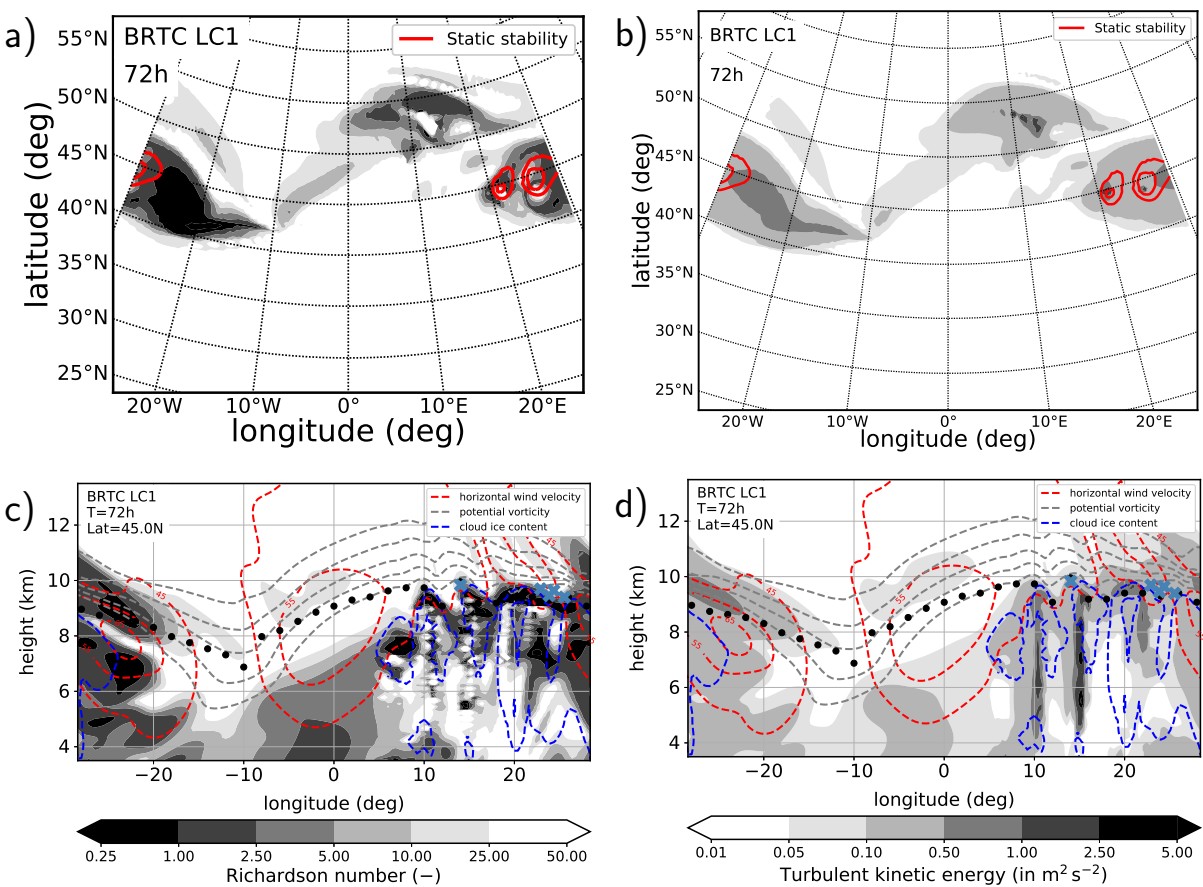

**Figure 13.** Left panels a) and c) show gradient Richardson number, $Ri$, right panels b) and d) TKE after 72 h of model integration. Upper panels show a horizontal cross-sectionw at the dynamic tropopause along with isolines of static stability with values of $N^2 = 5.5 \times 10^{-4}\,\text{s}^{-2}$ and $N^2 = 7.0 \times 10^{-4}\,\text{s}^{-2}$, lower panels show zonal cross sections at $45°$N in the center of the model domain, including isolines of various variables (see detailed description in Figure 11).



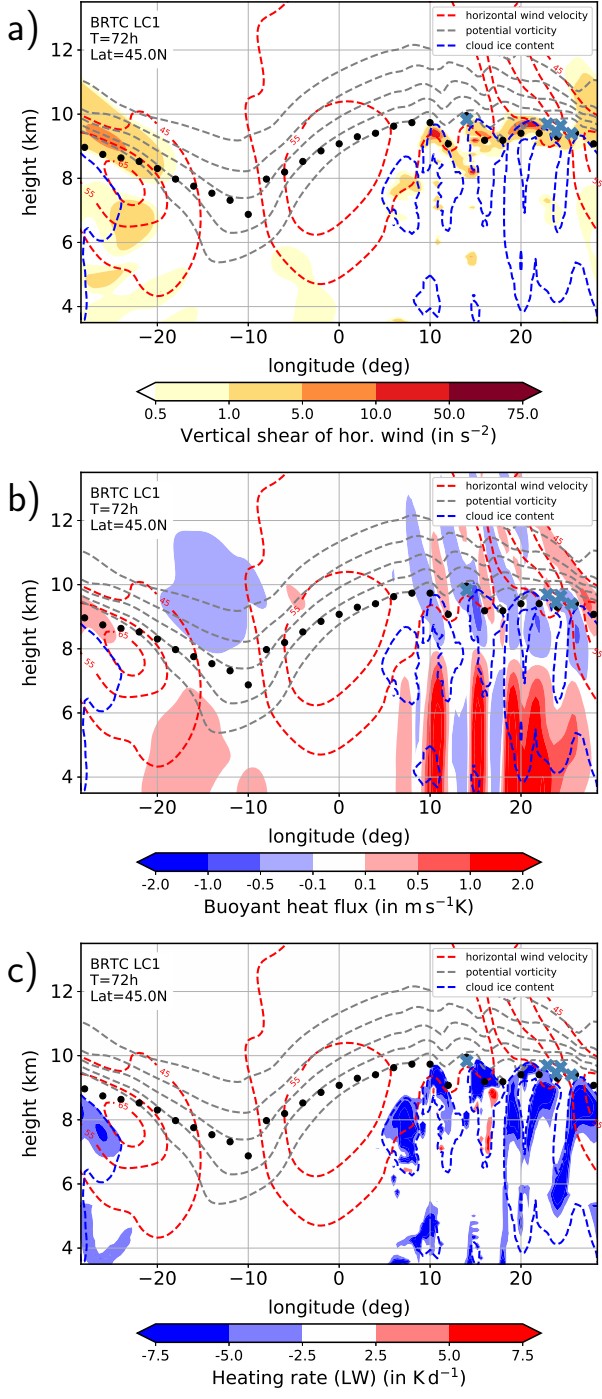

**Figure 14.** Zonal cross sections of a) vertical shear of horizontal wind $S^2$, b) buoyant heat flux $w'\Theta'_v$, and c) long wave heating rate $D\Theta_{LW}/Dt$ at 45 °N and after 72 h of model integration. For the description of isolines it is referred to Figure 11.





enhanced values of TKE (Figure 13). The Richardson number is the ratio between static stability and vertical shear of the horizontal wind. Since static stability shows rather large values above the tropopause, the source of low Richardson numbers results from large vertical shear of the horizontal wind (Figure 14a). The shear is largest in the ridge where a gravity wave is evident at the edges of the jet-stream. The shear also contributes to enhanced values of TKE and might thus explain its enhancement.

However, TKE further depends on the vertical gradient of the total moisture which can consequently lead to a buoyant heat flux (Doms, 2011). This buoyant heat flux shows positive and negative values at the tropopause, most probably related to the propagating and potentially dissipating gravity wave (Figure 14b). In particular, in the region of the TST trajectories negative values are dominant which could indicate upward motions in an otherwise stable environment. The stability is further increased by a radiative feedback of the ice clouds (Figure 14c). Coolling is evident at the top of the clouds which in turn enhances the

static stability above the tropopause in the region of the mixing (e.g., Fusina and Spichtinger, 2010; Kunkel et al., 2016). Thus, we identified the processes which allow for mixing induced by shear in a region which is stably stratified.

## 5  Summary and conclusions

A recent study by Kunkel et al. (2016) showed the concurrent occurrence of enhanced static stability in the lower stratosphere and increased turbulent motions across the extratropical tropopause in the ridge of idealized baroclinic life cycles. Here, we

present evidence that such a situation corresponds with mixing of trace gases at the level of or slightly above the tropopause and eventually with transport of tropospheric air into the stratosphere. To the authors' knowledge this process has gained only little attention so far, in particular in terms of the formation of the extratropical transition layer. We derive our conclusions from airborne measurements along with high resolution ECMWF model data to identify the occurrence of mixing and potential TST in the ridge of a baroclinic wave. We further extended experiments of idealized baroclinic life cycles from Kunkel et al. (2016)

to elucidate the driving mechanisms and to assess a more general picture of the mixing process.

During WISE RF07 signs of turbulent mixing are evident between 335-340 K potential temperature in the lowermost stratosphere just above the tropopause. The region of interest was situated above an extended cloud deck associated with a warm conveyor belt outflow and was substantially affected by small scale waves in the lowermost stratosphere. Vertical profiles of

tracers with a sufficient long life time in the UTLS such as $CO$ and $N_2O$ and their correlations are used to identify quasi-isentropic mixing between air masses of different origin. In particular, the $N_2O$ tracer mixing ratios suggest that tropospheric and stratospheric air masses mix in this region. Power spectral densities of potential temperature support the analysis suggesting that rather meso- than synoptic scale processes affect the power spectrum in the region of mixing.

Furthermore, ECMWF forecast data with the highest available resolution are used to obtain a broader view on the synoptic situation. Although the model performs well in representing the overall situation, some deviations have been found between model and measurements. In particular, these deviations occur at small scales which seem to be substantial in the representation of the mixing process. Nevertheless, trajectory calculations based on the model data show that mixing occurs around



the tropopause and affects the lower part of the extratropical transition layer. The mixing occurs in regions of enhanced lower stratospheric static stability, relative high potential temperature, anti-cyclonic flow, and during a time when the flow is affected by vertical shear, deformation, and moderate alternating vertical motions.

Moreover, numerical experiments of idealized baroclinic life cycles support the observational findings. In the model simulations it is evident that mixing occurs in the ridge of baroclinic waves, just above the top of ice clouds. Using special tracers and kinematic trajectories, we showed that even TST occurs in regions of enhanced static stability, thus in the region of the tropopause inversion layer. To a large degree the mixing is the result of a Kelvin-Helmholtz instability which has its source in enhanced vertical shear as has recently been demonstrated by Kaluza et al. (2018). This shear is strongly related to inertia-
gravity wave dynamics at the upper edge of the jet-stream. Moreover, buoyant heat fluxes caused by the upper-tropospheric clouds may also enhance the turbulence across the tropopause.

The overall relevance of this process needs yet to be analyzed. This is beyond of the current study. However, on the one hand this process occurs on small scales, and is related to cirrus clouds at the tropopause and a strong wind shear which in turn seems
to be related to small scale waves in the lower stratosphere. Thus, the impact on the trace gases in the upper troposphere and lower stratosphere might be not particularly large due to the limited geographical extent of the combinations of these processes. On the other hand all this occurs within baroclinic life cycles which in turn are relatively frequent features of the extratropical UTLS. Thus, there is potentially a non-negligible contribution of this mixing process on the composition of the extratropical transition layer. In particular, since this mixing occurs at relatively high potential temperatures, it can affect regions which
have previously not been considered to be strongly affected by STE in baroclinic waves. Previous studies showed that the main exchange in the extratropics occurs at lower potential temperature levels at the lower edge of the jet-stream.

One reason why the process described in this study has not gained much attention is that numerical weather prediction models and in particular reanalysis products as well as climate models do not resolve the UTLS sufficiently, thus potentially
miss or misrepresent the relevant processes. However, especially reanalysis data sets build the basis for almost all recent climatological studies of STE (e.g., Škerlak et al., 2014; Boothe and Homeyer, 2017). Figure 15 shows the same cross section for different ECMWF products. While the forecast shows many fine scale features, e.g., in the cloud structure and the location of the tropopause, these features are almost entirely missing in ERA Interim. This is to a large degree caused by the poorer vertical resolution of ERA Interim which is similar to the resolution in current climate models. In contrast, the new ERA5
reanalysis data which uses the same vertical grid spacing as the current forecast model shows more similarity to the forecast and might thus allow for a better representation of STE in the extratropics than ERA Interim. However, potentially even the forecast data still has problems to fully capture all features correctly, because the vertical grid spacing of that data is still relatively coarse compared to our idealized simulations with a vertical grid spacing of 110 m and a horizontal grid spacing of 0.4° which captured the TST relatively well. Thus, an increase of vertical model resolution seems to be necessary to further
address this process and its potential consequences also on the larger scale.





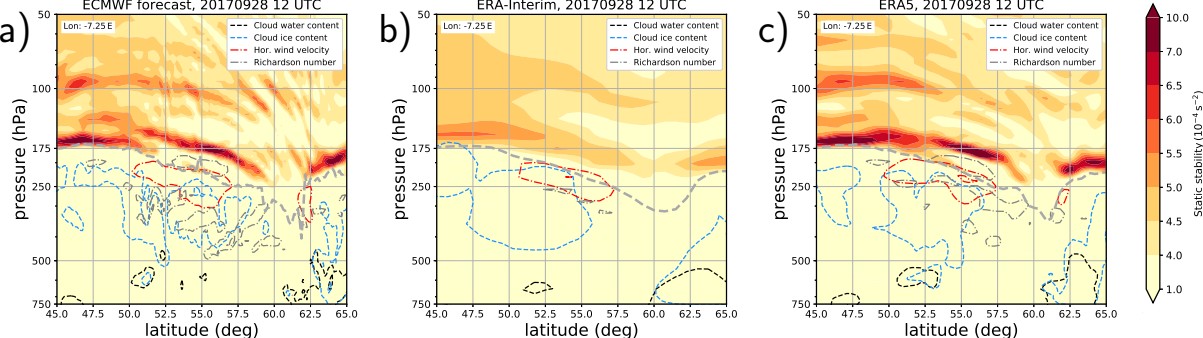

**Figure 15.** Zonal cross sections along $-7.25\,^\circ$E at 28.09.2017 12 UTC show static stability (color shaded), the altitude of the 2 pvu isosurface (light grey), cloud water content (black), cloud ice content (blue), horizontal wind speed (red), and Richardson number (dark grey). The three panels show a) the ECMWF forecast (with $0.07^\circ$ horizontal grid spacing and $\sim 300$ m vertical grid point spacing in the UTLS), b) the ERA Interim reanalysis (with $0.75^\circ$ horizontal grid spacing and $\sim 1000$ m vertical grid spacing), and c) the ERA5 reanalysis (with $0.25^\circ$ horizontal grid spacing and $\sim 300$ m vertical grid spacing).

*Code and data availability.* ECMWF (forecast, ERA Interim and ERA5) data has been retrieved from the MARS server. The airborne measurement data from the WISE campaign are available through the HALO data base (https://halo-db.pa.op.dlr.de/). The code of the COSMO model is available on request from the COSMO consortium (http://www.cosmo-model.org/). LAGRANTO is available from http://iacweb.ethz.ch/staff//sprenger/lagranto/. Output from the idealized COSMO simulations is available upon request (dkunkel@uni-mainz.de).

*Author contributions.* DK and PH designed the study. MR, MK, PH, DK organized the WISE campaign and were part of the scientfic flight planning team. DK, PH, BK analyzed the in situ data from WISE and JU provided GLORIA data. DK and TK analyzed ECMWF model data. DK ran the idealized simulations and analyzed the data with input from PH. DK wrote the paper with input from PH and TK; all authors contributed to the manuscript.

*Competing interests.* No competing interests.

*Acknowledgements.* Special thanks to the entire WISE team for the successful campaign. Logistics were handled by DLR-FX; many thanks for the great support and organization before, during, and after the campaign. Also a special thanks to the pilots for the realization of the specific flight patterns. More information about the WISE campaign can be found at https://www.wise2017.de/. We further thank all members of the GLORIA instrument team for their large efforts in developing the first airborne IR limb imager. The GLORIA hardware was mainly funded by the Helmholtz Association of German Research Centres through several large investment funds. The authors also thank H.



Wernli and M. Sprenger for the possibility to use LAGRANTO for this study. The authors are grateful to ECMWF for providing operational analysis and forecast as well as reanalysis data through the MARS server. The Authors acknowledge funding from the German Science Foundation, as this study was carried out as part of the preparation phase for the WISE campaign under funding from the HALO SPP 1294 (DFG grant no. KU 3524/1-1, HO 4225/7-1 as well HO 4225/8-1). Parts of this research were conducted using the supercomputer Mogon
5 and advisory services offered by Johannes Gutenberg University Mainz (hpc.uni-mainz.de), which is a member of the AHRP (Alliance for High Performance Computing in Rhineland Palatinate, www.ahrp.info) and the Gauss Alliance e.V. The authors gratefully acknowledge the computing time granted on the supercomputer Mogon at Johannes Gutenberg University Mainz (hpc.uni-mainz.de).



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
