# Peer review of "Evidence of small-scale quasi-isentropic mixing in ridges of extratropical baroclinic waves"

_Atmospheric Chemistry and Physics, 2019_

## Referee Comment (RC1) · Anonymous Referee #1 · 29 May 2019

**Referee report of the manuscript "*Evidence of small-scale quasi-isentropic mixing in ridges of extra-tropical baroclinic waves*" by Daniel Kunkel et al.**

This study presents a strong argument for the occurrence of quasi-isentropic turbulence-driven mixing or air masses near the tropopause in ridges of baroclinic waves. The evidence comes from a meticulous analysis of constituent measurements from the WISE campaign accompanied by meteorological data from ECMWF forecasts and by trajectory calculations. While the authors cannot quantify the overall effect of this process on the chemical composition of the LS using data from a single event (nor is it the goal of the study) the great value of this paper is in introducing this, to my knowledge, largely neglected mechanism of stratosphere-troposphere exchange. The authors nicely connect their analysis with previous results regarding the TIL associated with baroclinic waves and come to a neat and well-argued, perhaps surprising conclusion that the enhancements of static stability at the top of a ridge can enhance, rather than inhibit, mixing.

The paper is very well written, the analysis is thorough and convincing. It's one of those manuscripts that seem to anticipate the reader's questions and answer them. And I really like the photo in Fig. 2! I think that the study is suitable for publication in ACP almost as is. I only have very few, mainly technical comments.

**Minor comments**

P2. L14-17. I'm confused about the terminology here. In my reading of, e.g. Birner and Bönish 2011 and Abalos et al. 2017 the distinction between the shallow and deep branches of the Brewer-Dobson circulation is not identical with a separation between mean advection in mixing. Rather, the shallow branch transport is comprised of both two-way eddy mixing and slower advection by the residual circulation, the two largely balancing each other. The definitions of the two branches have more to do with transit times, stratospheric entry regions, etc. Am I wrong on this? It may be true that it is eddy mixing that connects the tropics with the high latitudes (as in Krause et al.) but in the region considered in the present manuscript, with latitudes south of 60N it seems the mean advection also plays a role. All this is tangential to the subject of this study and all I mean is that the terminology used here doesn't seem consistent with the literature. Also, I think that Birner and Bönish 2011 could be cited here.

P6 L13. The description of the COSMO model could use just a little more detail. It's a regional model, right? Where do the boundary conditions come from? Also, define the acronym (I believe, **Co**nsortium for **S**mall-scale **Mo**deling)

P16 L15. I think it should be stated explicitly what this classical meaning is. This comes up again below in Section 3.3.

P19. L14-17. Again, I'm confused about the "classical" STE and how it's opposed to what we have here. Isn't "an air parcel crossing the dynamic tropopause" the one and only meaning of TST? Does this sentence simply mean to say that we can't tell, based on the trajectory analysis, whether a STE event has occurred or not but the analysis provides evidence that it has? Maybe

it's just a matter of defining things more clearly. Also, I would say "classical sense" instead of "classical meaning"; just a preference.

**Technical corrections**

P4 L11. The acronym 'TIL' is introduced here but later "tropopause inversion layer" it is almost always spelled out throughout the paper. It should just be "TIL" from now on.

P3 L17. "All processes which lead to cross tropopause transport of air parcels have one common impact on this air parcel," There's something grammatically wrong with this sentence

P4 L14 initial → initially

P4 L22. "data of" → data from

P6 L4. forecast → forecast**s**

P7 L13, "in this study we use the 2 pvu isosurface as dynamic tropopause" this was already stated in the first paragraph of section 2.3. I suggest deleting this sentence.

P9 L5. "(Figures 1c,d)" Should it be 1c,e?

P10 L14-L16 This sentence a little awkward. It talks about crossing the tropopause above the tropopause, which doesn't make sense to me!

Fig 3a caption. Richardson number contour is not mentioned in the caption.

P10 L33. "defincies" → deficiencies(?)

P12 L5. "seek for" → seek (or search for)

Figure 4 Caption. I think the symbols $\Theta_M$ and $\Theta$ should be swapped. Also, lrt1 (first lapse rate tropopause) is not described in the caption or discussed in the text.

P20 L9. 240.000. I think you want a comma there: 240,000

P20 L13. propability → probability.

P23 L 10 passoing → passing

**References**

Abalos, M., W.J. Randel, D.E. Kinnison, and R.R. Garcia, 2017: Using the Artificial Tracer e90 to Examine Present and Future UTLS Tracer Transport in WACCM. *J. Atmos. Sci.,* **74**, 3383–3403, https://doi.org/10.1175/JAS-D-17-0135.1

Birner, T. and Bönisch, H.: Residual circulation trajectories and transit times into the extratropical lowermost stratosphere, Atmos. Chem. Phys., 11, 817-827, https://doi.org/10.5194/acp-11-817-2011, 2011.

---

## Referee Comment (RC2) · Anonymous Referee #2 · 6 Jun 2019

**Review for**

**Evidence of small-scale quasi-isentropic mixing in ridges of extra-tropical baroclinic waves**

**by Kunkel et al.**

**Summary:**

This paper addresses stratosphere-troposphere exchange (STE) that occurs in ridges of extratropical baroclinic waves. The topic is of interest to the readership of ACP, and the study stands out by considering both observations (aircraft measurements, analysis of stratospheric and tropospheric tracers and their correlation) and model results (ECMWF analyses, backward trajectories, idealized simulations). Observations and model results are combined into a coherent story why the location in ridges of baroclinic waves are particularly prone to STE. I definitely think that many interesting and relevant aspects are discussed to elucidate the physical processes at work. In particular, I like the trace gas analysis in figure 5. Still, there is also space for improvement. The main concerns are discussed below, and some minor points are then listed.

**Major concerns:**

**1. Dynamic vs. thermal tropopause:** Both definitions of the tropopause are used in the study, whereby I think that the authors are more inclined to the dynamic tropopause – which is OK. Still, I wonder why both definitions are needed for this study because the two definitions agree in a climatological sense, but locally the two tropopause heights can differ substantially. This, for instance, is the case where tropopause folds occur. I would therefore appreciate if the role of the two tropopause definitions is more clearly discussed. More specifically,

- P2,L32: Here it is written that 'these results are independent of the definition of the tropopause'. What exactly is meant by 'these results'?

- Figure 1 and corresponding text: In this figure, the local thermal tropopause is shown; but the dynamic tropopause is missing? Why? Actually, in the text (P7,L32) the measurement are discussed with respect to the height relative to the tropopause, without explicitly mentioning which definition of the tropopause is used (the thermal one; see figure 1) and 'misleading' the reader that the dynamic tropopause is used by mentioning the 'stratospheric PV' values (P7,L28), i.e. the key aspect of the dynamic tropopause.

- Figure 4 contains, in contrast, the dynamic and the thermal tropopause. Why?

- P21,L6-7: "We also find a spatial coincidence in the horizontal plane between the enhancement of N2 above the thermal tropopause and TST across the dynamic tropopause by analyzing passive tracers in our idealized simulations (Figure 10)." Here, both definitions of the tropopause are referred to. This is somewhat 'confusing' to me.

**2. Streamlining the introduction:** The introduction basically 'offers' everything that is needed for the study. But at some places I felt that a clear storyline was missing. Let me show this with some very specific examples:

- P2,L18 an L26: At both places it is written where STE occurs predominantly, i.e. it looks a little repetitive and the reader must read twice in which sense the two paragraphs differ.
- P3,L3-15: This paragraph discusses the crucial role of Rossby waves for STE, which I fully agree with. What I am missing is the link to the previous paragraph! As a reader I had the impression that this paragraph opens up a new story (Rossby waves), and does not 'naturally' evolve from the previous paragraph. Of course, this always reflects some personal view, but I think the introduction would benefit a lot if the story more clearly from one paragraph to the next.

Hence, many processes (radiation, folds, clouds, convection, gravity waves) are introduced in the introduction, but they remain somewhat 'unrelated' to the main topic (ridges in baroclinic waves). To be sure, I think it is fine to mention all these different aspects, but the processes should be streamlined (or directed) towards the topic of the paper.

**3. Role of the enhanced stability above the tropopause:** The storyline of the study is built around the enhanced stability near the extratropical tropopause and that STE is encountered in this region. While reading the text, I had the impression that a special role is attributed to this enhanced stability for STE. But there are several other processes at work: gravity waves (as discussed several times), turbulent regions due to lowered Richardson numbers.
In short, I wonder whether the whole story could also be interpreted in a different way, i.e. we encounter STE not because of the enhanced stability but despite of it. Then, the argument could be as follows: (i) a gravity wave evolves near or at the tropopause; (ii) because of this gravity wave vertical wind shears are increased and therefore the Richardson number becomes small; (iii) this reduction in the Richardson number due to the wind shear dominates any impact of the enhanced vertical stability and therefore leads to turbulent mixing and hence STE.
I don't know whether this is a valid interpretation of the current case, but it would see the enhanced vertical stability in a completely different light. I think the authors should discuss these alternative interpretations, or at least make clearer why the enhanced stability is so important for the mixing.

**4. Power spectral densities (Figure 6 and corresponding text):** The power spectrum is discussed in Figure 6 to show that isotropic turbulence (k=-5/3) prevails for flight leg (FL380), but that geostrophic turbulence (k=-3) prevails at later flight legs. The discussion should be clearer and in particular, I would lile the following aspects to be addressed:

- Why is it possible to identify structures down to 100 m with a sampling frequency of 2-3 Hz? Is this simply given by the aircraft's speed and the sampling period?

- Is there a reference that a slope of k=-3 is typical for gravity waves, as stated in the text (P15,L14-15)? I am certainly not an expert on power spectra, but I would have expected geostrophic turbulence to be typical at larger scales?

- The slope k=-3 (red line) seems to apply for a range between 0.01 and 0.2 Hz, whereas for smaller and larger frequencies it clearly deviates from this behavior (in figure 6b). How has this to be interpreted?

Basically, I think it is nice to have the power spectra in the manuscript, but I would appreciate a more detailed discussion.

**5. Idealized simulations:** In section 4, the authors refer to an idealized baroclinic life cycle simulation in Kunkel et al. (2016), more specifically to the experiment BRTC LC1. Of course, I understand that not all details of this previous simulation can be given. However, I would appreciate as a reader if could read (and understand!) the current paper without having read Kunkel et al. (2016) -- simply because I could not remember. Hence, I think that the authors should include as much details from Kunkel et al. (2016) in the current study that it becomes understandable without the previous study, i.e. it becomes more or less self contained.

As a specific example, In P20,L17 it is stated that STE starts to occur slightly after the time of the first enhancement of N^2. But where exactly is this N^2 value determined? I might have missed it in the current text, or it might indeed have to be got from Kunkel et al. (2016).

**Minor comments:**

- P2,L5: "certain trace species" -> You might want to specify already at this place what trace species are meant.

- P2,L6-78: Would it make sense to give, in addition to the height range above and below the dynamical tropopause, also in hPa or m?

- P2,L13: "in the deep branch into the UTLS" -> It is not immediately clear by the term 'deep branch', in particular if a reader is not very familiar with STE. It might be helpful to introduce in 1-2 sentences the stratospheric circulation with the distinct branches.

- P2,L16: "two competing transport pathways" -> Why are the two pathways competing? In which sense are they competing?

- P3,L19-20: "Lamarque and Hess (1994) separated between diabatic, i.e., potential temperature changing, and diffusive, i.e., related to friction, processes and showed that diabatic processes play a more vital role for STE than diffusive processes." -> Are there newer studies showing that diabatic processes than diffusive ones? I wonder whether this applies to STT and TST, and I am really not concvinced that turbulent mixing is less important (in particular for STT)?

- P3,L24: "Clouds and related diabatic heating" -> What are the diabatic heating processes related to clouds? Does it refer in particular to condensational heating (phase changes of water and ice)? Or is radiative cooling at cloud top also relevant?

- P3,L26-27: "... can reach the upper troposphere and modify the PV, consequently allowing for exchange between tropospheric and stratospheric air..." -> Note, however, that WCB air masses do not necessarily enter the stratosphere; the diabatic heating during the ascent is associated with mid-tropospheric PV changes, and the WCB is also able to modify the upper-level PV, but further diabatic and/or diffusive processes are needed that the air masses cross the tropopause.

- P4.L31+33: "took place" & "are to examine"; the tense is switching from past to present; please make this consistent (not only at this place).

- P5,L33-34: "had the goal to study the abundance of trace species in the extratropical tropopaus region in relation to the occurrence of enhanced values of static stability in the lower stratosphere and to potential STE." -> Please rephrase in a clearer way; as a suggestion: ".. in the extratropical tropopause region and how they are influenced by the enhanced static stability ... and potential STE. Such conditions were found by Kunkel et al. (2016) to occur in the ridges of extratropical baroclinic waves. Therefore,..."

- P6,L9: Why is a slightly degraded horizontal grid (0.125 deg) for the trajectory calculation compared to the other analysis (0.07 deg)?

- P7,L7: The horizontal resolution of the COSMO output is 0.4 deg; in contrast, it is 0.125 deg for ECMWF, i.e. it is higher for ECMWF than for COSMO. Is this correct? How do the vertical spacing of ECMWF and COSMO compare in the UTLS?

- Section: 3.1: Would it be possible to have one figure (or figure panel) where all flight legs are labeled? While reading this section it was difficult to immediately know where the flight legs are. For instance, it would help to locate the flight legs of figure 3 more easily.

- P10,L23-: Here, a model deficiency is discussed, namely the too high values of the Richardson number near some regions around the tropopause. This discussion of a model deficiency somewhat interrupts the main storyline, and hence I wonder whether it should better be discussed in section 2.2 where ECMWF data are introduced? Furthermore, the term 'in some regions' is rather unspecific, and immediately lets the reader ask where these regions are.

- P10,L34-35: "Thus, the model forecast underestimates the strength of the inversion, most potentially due to definces in representing the gravity wave in this region." -> First, note the spelling error! Then, how sure are you that this is indeed a gravity wave? Then, the underestimation of the strength of the inversion is attributed to the effect of the gravity waves, i.e. because they are not well enough captured by the model. How do you know that this underestimation is not because of a limited vertical (and horizontal) resolution of the model?

- P13,L2-3: "In general, at the tropopause the CO–N2O correlation starts with larger CO and larger N2O mixing ratios at potential temperatures typical for the extratropical tropopause" -> 'larger' refers to a comparison; but to what is it compared? Of course, I see the point, but I think it is not perfectly clear. Furthermore, I wonder whether it is correct to say that a correlation starts at a larger N2O and CO values. It sounds a little strange to me!

- P16,L12: What does "at some time" mean? Or, stated otherwise: How long are the backward trajectories? Possibly, I simply missed this piece of information, and if not: Please add it!

- P16,L23.24: " Starting from this region the trajectories strongly decelerate in a region of alternating horizontal divergence. During this time the trajectories cross back and forth over the dynamic tropopause." -> How do you see in figure 7 that the trajectories are decelerating? How do you interpret the alternating horizontal divergence? Is the divergence pattern due to the gravity wave?

- P16,L26-27: " The motion back and forth across the chosen PV value for the dynamic tropopause becomes also evident from the PV along the trajectories (Figure 8a)." -> I am not sure whether I see this crossing back and forth over the dynamic tropopause in figure 8. What I see is that both sides (PV smaller and larger than 2 PVU) are 'covered' by the trajectories, but no further details.

- Figure 7: In the text it is written that a transition (although not a smooth one) can be seen in the PV (panel b). This is difficult to see in my print out. It might also be helpful to have the relevant flight legs added to the figure; otherwise, it is a little 'difficult' to relate the trajectories to the measurements.

- Figure 8: Why does the scale go up to 10 PVU in panel a)? In the same line, would it be possibly to adjust the scale in panel d)? Here, Ellrod and Knapp's TI index is shown as an additional turbulence indicator? Earlier in the text, only Richardson number was considered? I think it would be nice to be consistent throughout the manuscript, i.e. either to discuss only one or both indices. Finally, would it make sense to zoom in into a shorter time period around 15 UTC? For example, from -8 h to + 8 h.

- P19,L3: Where does the number 69048 come from? Is this the starting frequency of the trajectories times the duration of the time period?

-P19,L14-23: Here, it is discussed whether the STE (and in particular TST) follows the classical meaning of TST. What is 'the classical meaning of TST'? The term is unclear! Actually, I wonder whether this whole discussion about 'classical' or 'not classical' is necessary? If the authors would like to keep it, a more detailed discussion about the exact meaning of this term has to be included, and it has to be made more clear why it is relevant for the study.

- Figure 9 and the corresponding text: This figure shows the maximum static stability in the idealized baroclinic life cycle experiment BRTC LC1. I have several questions qith respect to

this figure: (i) How robust is the maximum static stability? (ii) Are only STT and TST trajectories in the ridges of a baroclinic wave included? I guess that this is not the case, but if so: It distracts the reader from the main story, which is about STE at exactly these locations.

- P21,L3-4: "Thus, there is almost a temporal coincidence between the start of the enhancement of static stability and the first occurrence of STE." -> The statement is fine, but it repeats essentially the first sentence of the paragraph. Hence, it is somewhat repetitive!

- Figure 10: Where are these structures relative to the ridge and trough of the baroclinic wave?

- P22,L4-5: "In contrast to TST and although a temporal coincidence is also evident for N2 enhancement and occurrence of STT, no spatial co-occurrence is evident for STT in regions of enhanced N2 (Figures 10c,d)" -> Rephrase in a clearer way? How do you infer that STT does not co-occur with enhanced $N^2$? Is this statement based only on the blue area in figure 10? Wouldn't we need a stratospheric tracer to make such a statement?

- P22,L8: "just above a region of ice cloud occurrence" -> What is the relevance of these ice clouds?

- P22,L9: "wave pattern related to a propagating inertia gravity wave" -> The gravity wave seems to be rather important for the mixing across the tropopause? It is not completely clear to me where this gravity wave originates from? Further, it is written that the wave propagates? But in which direction? It seems to me, based on the vertical cross sections, that the wave does not really propagate in the vertical direction. Instead, could it be that the gravity wave actually propagates along the troposphere-stratosphere interface, i.e. along the tropopause? A more refined analysis would be very helpful, given that the wave patterns is mentioned at several places in the manuscript.

- P26,L8-9: "A common feature of the two sets of trajectories is that in both cases the potential temperature values hardly change in the six hours; thus, the TST occurs quasi-isentropically." -> How do you see this in the figure? Or is this statement based on a quantitative analysis of the trajectories? I think a more detailed discussion of the quasi-isentropic transport is necessary, in particular because this is one of the key words in the article's title.

---

## Author Comment (AC1) · 21 Aug 2019

**Reply to referee comment #01**

We appreciate the kind words on our manuscript and thank the reviewer for the constructive comments and proposed suggestions. These helped to substantially improve the manuscript. Please note that we changed figures 1, 2, 7, 8, 10, 12 based on comments of the reviewers. We also added an appendix in which we briefly describe the idealized model setup in more detail.
We will answer to all comments of reviewer #01 below point by point. Referee comments are given in bold, answers in standard, and changes to the manuscript in italic font.

**The paper is very well written, the analysis is thorough and convincing. It's one of those manuscripts that seem to anticipate the reader's questions and answer them. And I really like the photo in Fig. 2! I think that the study is suitable for publication in ACP almost as is. I only have very few, mainly technical comments.**

**Minor comments:**

> **P2. L14-17. I'm confused about the terminology here. In my reading of, e.g. Birner and Bönish 2011 and Abalos et al. 2017 the distinction between the shallow and deep branches of the Brewer-Dobson circulation is not identical with a separation between mean advection in mixing. Rather, the shallow branch transport is comprised of both two-way eddy mixing and slower advection by the residual circulation, the two largely balancing each other. The definitions of the two branches have more to do with transit times, stratospheric entry regions, etc. Am I wrong on this? It may be true that it is eddy mixing that connects the tropics with the high latitudes (as in Krause et al.) but in the region considered in the present manuscript, with latitudes south of 60N it seems the mean advection also plays a role. All this is tangential to the subject of this study and all I mean is that the terminology used here doesn't seem consistent with the literature. Also, I think that Birner and Bönish 2011 could be cited here.**

Thanks for pointing this out. Of course, the shallow branch consists not only of two-way eddy mixing but also advection by the residual circulation.
We changed the text here as follows also taking into account the comments of reviewer #2:

*"The stratospheric circulation contributes with two distinct branches distinguished based on the transit time between the major entry point of air into the stratosphere, i.e., the tropical tropopause, and the extratropics. The so called deep branch, i.e., associated with long transit times affects the ExTL through the descent of old stratospheric ozone rich air into the UTLS. In contrast, the shallow branch with significantly shorter transit times introduces relatively young air from the tropical and subtropical UTLS into the extratropical UTLS (Birner and Bönisch, 2011). A recent study based on airborne measurements showed the effect of these two transport pathways on the changing abundance of carbon monoxide (CO) over the course of the Arctic winter in the ExTL (Krause et al., 2018)."*

> **P6 L13. The description of the COSMO model could use just a little more detail. It's a regional model, right? Where do the boundary conditions come from? Also, define the acronym (I believe, Consortium for Small-scale**

**Modeling)**

We initially kept this description brief, because it is a repetition of Kunkel et al. (2014, 2016) and we wanted only to provide the most necessary details to follow the discussion later in the manuscript. However, since both reviewers ask for more details, we extended the description of the COSMO model setup in the revised version and added more details to the new Appendix A.

**P16 L15. I think it should be stated explicitly what this classical meaning is. This comes up again below in Section 3.3.**

At this point the classical meaning of TST is exactly what the reviewer suggests in the next comment, i.e., an air parcel irreversibly crossing the tropopause from the troposphere into the stratosphere (as indicated by changing PV). However, although we think that our measurements show that air parcels from the troposphere must have entered the stratosphere, the trajectories are not that clear. In our trajectory analysis we do not find a coherent ensemble of trajectories entering the stratosphere just once and then staying there. We rather find trajectories coming from the troposphere, thus carrying a tropospheric chemical signature, which cross the tropopause multiple times. Note, that Wernli and Bourqui (2002, their Figure 1) introduced a residence time criterion which accounts for such multiple changes. The question remains whether this back and forth is something that happens as such in nature or whether this is potentially indicative for a gradual transition between the troposphere and stratosphere. Another option is of course that this behavior is rather an artifact due to the inability of the model to represent this situation entirely correct. For further clarification we rephrase this paragraph as follows:

*"Thus, for further discussion we rather omit the terminology of TST and STT trajectories in a sense of coherent ensembles of trajectories which cross the tropopause only once from the troposphere (stratosphere) to the stratosphere (troposphere) (Wernli and Davies, 1997; Stohl et al., 2003). We rather think of trajectories which show the potential of mixing around the tropopause by encountering low Richardson numbers and having PV values changing between tropospheric and stratospheric values, nevertheless leading to a subsequent exchange across the tropopause."*

**P19. L14-17. Again, I'm confused about the "classical" STE and how it's opposed to what we have here. Isn't "an air parcel crossing the dynamic tropopause" the one and only meaning of TST? Does this sentence simply mean to say that we can't tell, based on the trajectory analysis, whether a STE event has occurred or not but the analysis provides evidence that it has? Maybe it's just a matter of defining things more clearly. Also, I would say "classical sense" instead of "classical meaning"; just a preference.**

Here and according to the comments above we change parts of the paragraph to:

*"However, based on the trajectory analysis it is difficult to estimate whether STE and in particular TST occurs in a way that air parcels cross the dynamic tropopause*

*only once from the troposphere into the stratosphere and stay there afterwards. We also performed longer trajectory calculations which, however, also provide no further evidence of TST trajectories which then stay in the stratosphere. We rather find trajectories which, based on PV, alternate back and forth between troposphere and stratosphere and which encounter low Richardson numbers along their paths. One potential reason why no TST trajectories are found which stay in the stratosphere is that the model fails to correctly resolve the process. "*

**Technical corrections**

**P4 L11. The acronym 'TIL' is introduced here but later "tropopause inversion layer" it is almost always spelled out throughout the paper. It should just be "TIL" from now on.**

We changed tropopause inversion layer to TIL on the following pages.

**P3 L17. "All processes which lead to cross tropopause transport of air parcels have one common impact on this air parcel," There's something grammatically wrong with this sentence**

We changed this sentence to:

*"An air parcel crossing the tropopause has to be affected by processes which can modify its potential vorticity (Hoskins et al., 1985). Only then the air parcel can enter from a region with generally low PV, i.e., the troposphere, into a region of high PV, i.e., the stratosphere or vice versa."*

**P4 L14 initial → initially**

Changed as suggested.

**P6 L4. forecast →forecasts**

Changed as suggest.

**P7 L13, "in this study we use the 2 pvu isosurface as dynamic tropopause" this was already stated in the first paragraph of section 2.3. I suggest deleting this sentence.**

We deleted this sentence.

**P9 L5. "(Figures 1c,d)" Should it be 1c,e?**

Correct. However, since we modified Figure 1, this part refers now to Figures 1b,c.

**P10 L14-L16 This sentence a little awkward. It talks about crossing the tropopause above the tropopause, which doesn't make sense to me!**

We changed this sentence to:

*"At FL 380 HALO was initially flying in the lowermost stratosphere, then gradually approaching and finally crossing the dynamic tropopause which was slightly tilted according to the ECMWF analysis (Figure 3a)."*

**Fig 3a caption. Richardson number contour is not mentioned in the caption.**

We added the description to the caption.

**P10 L33. "defincies" → deficiencies(?)**

Correct. We changed it accordingly.

**P12 L5. "seek for" → seek (or search for)**

Correct. We changed it accordingly.

**Figure 4 Caption. I think the symbols Q M and Q should be swapped. Also, lrt1 (first lapse rate tropopause) is not described in the caption or discussed in the text.**

Thanks for pointing this out. We corrected the symbols and we included the description of lrt1 to the figure caption. In the text it is now also referred to the lrt1. We also note that we kept the lrt1 because in this figure it nicely shows how close the lrt1 and dynamic tropopause are in this case.

**P20 L9. 240.000. I think you want a comma there: 240,000**

True. Changed it accordingly.

**P20 L13. Propability → probability.**

Correct. We changed it accordingly.

**P23 L 10 passoing → passing**
Correct. We changed it accordingly.

**References:**
Wernli, H. and Bourqui, M.: A Lagrangian "1-year climatology" of (deep) cross-tropopause exchange in the extratropical Northern Hemisphere, J. Geophys. Res., 107(D2), 4021, doi:10.1029/2001JD000812, 2002.

---

## Author Comment (AC2) · 21 Aug 2019

**Reply to referee comment #02**

We appreciate the careful reading of our manuscript by the reviewer and thank the reviewer for the constructive comments and proposed suggestions. These helped to substantially improve the manuscript. Please note that we changed figures 1, 2, 7, 8, 10, 12 based on comments of the reviewers. We also added an appendix in which we briefly describe the idealized model setup in more detail.
We will answer to all comments of reviewer #01 below point by point. Referee comments are given in bold, answers in standard, and changes to the manuscript in italic font.

**Major concerns**

1. **Dynamic vs. thermal tropopause: Both definitions of the tropopause are used in the study, whereby I think that the authors are more inclined to the dynamic tropopause – which is OK. Still, I wonder why both definitions are needed for this study because the two definitions agree in a climatological sense, but locally the two tropopause heights can differ substantially. This, for instance, is the case where tropopause folds occur. I would therefore appreciate if the role of the two tropopause definitions is more clearly discussed.**

   Thanks for pointing out this issue. Generally, as the reviewer noticed, in terms of STE we are more inclined to the dynamic tropopause, most specifically because its definition based on potential vorticity and the associated conservation law which to first order is often fulfilled close to the tropopause. On the other the hand, the static stability is a thermodynamic quantity related to the temperature of the atmosphere. More specifically, the enhanced values of static stability are associated with the temperature inversion which is commonly found above the lapse rate tropopause. One goal of WISE was to study whether STE occurs in the vicinity of enhanced static stability. From our point of view it is then necessary to have an eye on both the lapse rate and potential vorticity based tropopause in our discussion. However, to avoid further confusion and since the focus of the study is on mixing and potential STE we will generally refer to the dynamic tropopause in the manuscript and only use the lapse rate tropopause when necessary. This will then also be highlighted explicitly in the text.
   To clarify things more, we will add a sentence in the manuscript at the end of Section 2.4:

   *"Finally, note that in this study discussion around static stability are usually associated with the lapse rate tropopause, while discussion about STE are linked to the dynamic tropopause. Thus, both definitions will be considered in the analysis, however, if only the term tropopause is used, then we refer to the dynamic tropopause with a value of 2 pvu throughout the manuscript."*

   **More specifically,**
   **P2,L32: Here it is written that 'these results are independent of the definition of the tropopause'. What exactly is meant by 'these results'?**

   These results referred to the spatial and temporal occurrence of STE. We changed the text accordingly:

*"Also in a climatological sense the occurrence of STE in the extratropics is independent of the definition of the tropopause as shown by Boothe and Homeyer (2017) who used four different modern reanalysis data sets to analyze STE as well as lapse-rate and dynamic tropopause definitions."*

**Figure 1 and corresponding text: In this figure, the local thermal tropopause is shown; but the dynamic tropopause is missing? Why? Actually, in the text (P7,L32) the measurement are discussed with respect to the height relative to the tropopause, without explicitly mentioning which definition of the tropopause is used (the thermal one; see figure 1) and 'misleading' the reader that the dynamic tropopause is used by mentioning the 'stratospheric PV' values (P7,L28), i.e. the key aspect of the dynamic tropopause.**

The initial intention of Figure 01 was to give an overview of the synoptic situation. For this we decided to show PV and $N^2$ along with several quantities which from our point of view were relevant for our study. We also wanted to connect our story to the point where Kunkel et al. (2016) stopped their analysis with respect to the exchange. For this we started our discussion around STE (wrt to the dynamic tropopause) and enhanced stability (wrt to the lapse rate tropopause) to link these two features. However, since this apparently led to some confusion and to a too strong focus on the static stability, we decided to rearrange Figure 01. Now we mainly show the distribution of PV and start the discussion about static stability later in the text. Figure 03 also shows the most important features of static stability that were evident in the original Figures 01d,e. Static stability is only included as one of several contour lines in the revised Figure 01. Please note that the new Figure 1 has a slightly different color-scale for PV which has one value for negative PV values which occur for instance at the top of the WCB. Also tropospheric values (on the Northern Hemisphere and based on our dynamic tropopause definition) are now represented by one value/one color, so that the dynamic tropopause is more easily detectable than before. The color-scale then increases by 0.5 pvu from 2 pvu to 10.5 pvu to illustrate the high PV variability in the lowermost stratosphere. We hope that this and the general clarification on the term tropopause as described above reduces the confusion about tropopause definitions.

Figure 01: Revised Figure 01, now focusing on potential vorticity to show the synoptic situation

[Figure]

**Figure 4 contains, in contrast, the dynamic and the thermal tropopause. Why?**

Figure 4 shows time series of measured and modeled quantities, however, only along the flight path without any real two-dimensional information. The intention was to include both, lapse rate and dynamic tropopause, to show that in this case these two hardly differ. Since the TIL is directly linked to the lapse rate tropopause, the lrt1 can be further used as an indicator for the TIL location, while the dynamic tropopause gives a rough idea how deep HALO was flying in the stratosphere.

**P21,L6-7: "We also find a spatial coincidence in the horizontal plane between the enhancement of N2 above the thermal tropopause and TST across the dynamic tropopause by analyzing passive tracers in our idealized simulations (Figure 10)." Here, both definitions of the tropopause are referred to. This is somewhat 'confusing' to me.**

However, from our point of view it is necessary to show both, since STE is related to the dynamic and static stability rather to the lapse rate tropopause. As noted by the reviewer, thermal and dynamic tropopause can coincide but do not necessarily have to. This is for instance the case in the region of the trough. Following Figure 10 we included a cross section in Figure 11. There it is shown that the two tropopause definitions coincide in altitude at the location of the exchange. This further indicates that the diagnosed exchange does not strongly depend on the tropopause definition in the current study.

2. **Streamlining the introduction: The introduction basically 'offers' everything that is needed for the study. But at some places I felt that a clear storyline was missing. Let me show this with some very specific examples:**

   **P2,L18 an L26: At both places it is written where STE occurs predominantly, i.e. it looks a little repetitive and the reader must read twice in which sense the two paragraphs differ.**

   We revised some of the paragraphs focusing on STE in the introduction. In particular, we shortened paragraph 2 and include the most relevant information in the fourth paragraph.

   **P3,L3-15: This paragraph discusses the crucial role of Rossby waves for STE, which I fully agree with. What I am missing is the link to the previous paragraph! As a reader I had the impression that this paragraph opens up a new story (Rossby waves), and does not 'naturally' evolve from the previous paragraph. Of course, this always reflects some personal view, but I think the introduction would benefit a lot if the story more clearly from one paragraph to the next.**

   The intention was that the paragraph starting on P2, L24 briefly discusses climatological features of STE, while the following paragraph lists the STE relevant processes in the extratropics. We start this paragraph now also with the following sentence:

*"Within the storm tracks Rossby waves are crucial for STE."*

**Hence, many processes (radiation, folds, clouds, convection, gravity waves) are introduced in the introduction, but they remain somewhat 'unrelated' to the main topic (ridges in baroclinic waves). To be sure, I think it is fine to mention all these different aspects, but the processes should be streamlined (or directed) towards the topic of the paper.**

We agree that the introduction is comprehensive, in particular with the goal to cover all processes related to baroclinic waves which potentially lead to STE. We also wanted to make sure that although STE in the extratropics has been addressed quite comprehensively in the literature, we found a process which has not been discussed as it is done in the current study..

3. **Role of the enhanced stability above the tropopause: The storyline of the study is built around the enhanced stability near the extratropical tropopause and that STE is encountered in this region. While reading the text, I had the impression that a special role is attributed to this enhanced stability for STE. But there are several other processes at work: gravity waves (as discussed several times), turbulent regions due to lowered Richardson numbers.**
**In short, I wonder whether the whole story could also be interpreted in a different way, i.e. we encounter STE not because of the enhanced stability but despite of it. Then, the argument could be as follows: (i) a gravity wave evolves near or at the tropopause; (ii) because of this gravity wave vertical wind shears are increased and therefore the Richardson number becomes small; (iii) this reduction in the Richardson number due to the wind shear dominates any impact of the enhanced vertical stability and therefore leads to turbulent mixing and hence STE.**
**I don't know whether this is a valid interpretation of the current case, but it would see the enhanced vertical stability in a completely different light. I think the authors should discuss these alternative interpretations, or at least make clearer why the enhanced stability is so important for the mixing.**

Our intention was indeed to show that mixing occurs despite the enhanced static stability and not because of it. The line of argumentation outlined by the reviewer is fully valid and also follows the line of argumentation in the paper:
i) a gravity wave is evident above the tropopause (Fig.3)
ii) increased vertical wind shear and related reduction in Richardson number (Fig. 4)
iii) the increased wind shear outweighs even the enhanced static stability found in the lower stratosphere in the regions of mixing (Fig 3, 4)
Figure 5 shows observed indications of mixing. Figure 6 shows that small scale dynamic features seem to dominate the flow (i.e., the presence of the small scale waves) while Figure 7 and 8 extent the analysis and shed light onto the synoptic situation and a bit on the ability of the ECMWF model and kinematic trajectory models running on this data to capture this process.
Section 4 was then designed to i) further analyze and generalize the findings presented in Section 3 and ii) extend the analysis of Kunkel et al. (2016).
As stated above in the revised manuscript we changed Figure 01 to reduce the initial focus on static stability.

Another point is the question that is placed in the 7th paragraph of the introduction (P4, L12). We change this question as follows:

*"Does STE occur and does it affect the formation of the ExTL in the ridge of baroclinic waves where the static stability is usually strong in the extratropical lowermost stratosphere?"*

4. **Power spectral densities (Figure 6 and corresponding text): The power spectrum is discussed in Figure 6 to show that isotropic turbulence (k=-5/3) prevails for flight leg (FL380), but that geostrophic turbulence (k=-3) prevails at later flight legs. The discussion should be clearer and in particular, I would like the following aspects to be addressed:**
   **Why is it possible to identify structures down to 100 m with a sampling frequency of 2-3 Hz? Is this simply given by the aircraft's speed and the sampling period?**

   The 100 m rather correspond to the 10 Hz data provided by the BAHAMAS instrument of HALO, i.e. for temperature and wind. Along with an average ground speed of about 210 ms$^{-1}$, this would allow us to identify structures down to 100 m, taken into account that several independent measurement points are required to identify a structure. Of course, the trace gas measurements with about 3 Hz would not allow to identify such small structures.
   We changed the sentence on P15 L6-7 to make this clearer:

   *"On board HALO the frequency of measurements is about 10 Hz for the state parameters such as temperature and wind, while it is about 2-3 Hz for CO and N$_2$O. Along with an average ground speed of about 210 ms$^{-1}$ and with the 10 Hz measurements of the state parameters this potentially allows us to identify atmospheric structures down to about 100 m."*

   **Is there a reference that a slope of k=-3 is typical for gravity waves, as stated in the text (P15,L14-15)? I am certainly not an expert on power spectra, but I would have expected geostrophic turbulence to be typical at larger scales?**

   This last sentence is thought as a summary of this paragraph, summarizing the findings of the paragraph. As outlined in the text (P15, L10) a slope of k=-5/3 is thought to be characteristic for isotropic turbulence and k=-3 for geostrophic turbulence (P15, L13).
   We rephrased the last sentence to make it clearer that this sentence is thought as a summary of the power spectral density discussion:

   *"Summarizing, close to the tropopause the slope of k=-5/3 suggests that meso-scale processes, e.g., related to gravity waves, might be substantial to explain the dynamics in the tropopause region."*

   **The slope k=-3 (red line) seems to apply for a range between 0.01 and 0.2 Hz,whereas for smaller and larger frequencies it clearly deviates from this behavior (in figure 6b). How has this to be interpreted?**

The slope of k=-3 is evident to about 1 Hz, if it is taken into account that there is a sort of offset at around 0.2 Hz. Afterwards the spectral density flats out, which could be sign of simply missing meso-scale features while other larger scale features become more relevant again which are, however, not fully resolved by our measurements.

**Basically, I think it is nice to have the power spectra in the manuscript, but I would appreciate a more detailed discussion.**

We appreciate the interest in the power spectral density. Our intention was to show that turbulent motions occurred in the vicinity of the tropopause which have their sources in processes on the meso scale. In contrast, these processes seem to have no relevance farther away from the tropopause. The power spectral densities in Figure 6 nicely show this behaviour for the two considered legs (FL380 and FL420). Of course, the power spectral densities allow for a much deeper analysis, which is however, beyond the scope of the current manuscript but is the focus on current ongoing analyses which will then cover the power spectral densities in more detail.

5. **Idealized simulations: In section 4, the authors refer to an idealized baroclinic life cycle simulation in Kunkel et al. (2016), more specifically to the experiment BRTC LC1. Of course, I understand that not all details of this previous simulation can be given. However, I would appreciate as a reader if could read (and understand!) the current paper without having read Kunkel et al. (2016) -- simply because I could not remember. Hence, I think that the authors should include as much details from Kunkel et al. (2016) in the current study that it becomes understandable without the previous study, i.e. it becomes more or less self contained.**
**As a specific example, In P20,L17 it is stated that STE starts to occur slightly after the time of the first enhancement of N^2. But where exactly is this N^2 value determined? I might have missed it in the current text, or it might indeed have to be got from Kunkel et al. (2016).**

We understand that the discussion of the idealized life cycles is probably too short in the current manuscript since this issue has been addressed by both reviewers. We thus decided to give a more comprehensive description of the model setup in an appendix. We also added some more information on how the $N^2_{max}$ is traced during the barolinic life cycle experiment to the caption of Figure 9.

**Minor comments:**
**P2,L5: "certain trace species" -> You might want to specify already at this place what trace species are meant.**

We change this to:

*"...distributions of trace species such as CO, $O_3$, $H_2O$, or $N_2O$ with either distinct tropospheric or stratospheric…."*

**P2,L6-78: Would it make sense to give, in addition to the height range above and below the dynamical tropopause, also in hPa or m?**

Hoor et al. (2004) does not give a height range in hPa or m, from Hegglin et al. (2009) the top of the ExTL was estimated to be about 2 km above the dynamical tropopause based on CO-O$_3$ correlations. However, there is no fixed relation between geometric altitude and potential temperature, this depends rather on the time and space varying vertical gradient of potential temperature. This in turn is highly dependent on the synoptic situation. Thus, any numbers in geometric space are relatively vague and are associated with larger uncertainty (see Hoor et al. (2004) and Hegglin et al. (2009)). Because of this we rather keep the initial formulation, only giving the range for potential temperature.

**P2,L13: "in the deep branch into the UTLS" -> It is not immediately clear by the term 'deep branch', in particular if a reader is not very familiar with STE. It might be helpful to introduce in 1-2 sentences the stratospheric circulation with the distinct branches.**

Following the suggestions of both reviewers we changed this paragraph accordingly:

"*The stratospheric circulation contributes with two distinct branches distinguished based on the transit time between the major entry point of air into the stratosphere, i.e., the tropical tropopause, and the extratropics. The so called deep branch, i.e., associated with long transit times affects the ExTL through the descent of old stratospheric ozone rich air into the UTLS. In contrast, the shallow branch with significantly shorter transit times introduces relatively young air from the tropical and subtropical UTLS into the extratropical UTLS (Birner and Bönisch, 2011). A recent study based on airborne measurements showed the effect of these two transport pathways on the changing abundance of carbon monoxide (CO) over the course of the Arctic winter in the ExTL (Krause et al., 2018).*"

**P2,L16: "two competing transport pathways" -> Why are the two pathways competing? In which sense are they competing?**

See last comment, we removed the competing.

**P3,L19-20: "Lamarque and Hess (1994) separated between diabatic, i.e., potential temperature changing, and diffusive, i.e., related to friction, processes and showed that diabatic processes play a more vital role for STE than diffusive processes." -> Are there newer studies showing that diabatic processes than diffusive ones? I wonder whether this applies to STT and TST, and I am really not convinced that turbulent mixing is less important (in particular for STT)?**

We fully agree that turbulence might have been underestimated in older studies. A potential explanation is that processes leading to turbulence in the UTLS have been poorly treated in models in these studies. An example is that Gray (2006) found similar results as Lamarque and Hess (1994) with cloud and radiative diabatic effects dominating over turbulent processes with respect to STE. In contrast, a recent study by Spreitzer et al. (2019) who by applying a Lagrangian technique could show that turbulence plays a major role in changing PV around the tropopause in the ridge of an extratropical cyclone and that cloud related diabatic

processes are more relevant at lower tropopause altitudes. We extent the discussion here as follows:

*"Lamarque and Hess (1994) separated between diabatic, i.e., potential temperature changing, and diffusive, i.e., related to friction, processes and showed that diabatic processes play a more vital role for STE than diffusive processes. Analyzing cross tropopause transport in the UKMO model Gray (2006) found similar results with cloud and radiative processes being more important for STE than processes related to turbulence. In contrast, a recent study by Spreitzer et al. (2019) shows that turbulent processes are mainly responsible for changing the PV around the tropopause in a ridge of an extratropical baroclinic wave. They used high resolution ECMWF forecast data and conclude that the turbulence is evident around the jet stream. This turbulence is mainly related to the vertical shear of the jetstream but can also be caused by gravity waves (Zhang et al. 2015a)."*

**P3,L24: "Clouds and related diabatic heating" -> What are the diabatic heating processes related to clouds? Does it refer in particular to condensational heating (phase changes of water and ice)? Or is radiative cooling at cloud top also relevant?**

With "clouds and related diabatic heating" we refer to all diabatic processes related to clouds, that those triggered by phase transitions (condensation, evaporation, etc) but also radiative processes related to clouds, e.g., radiative cooling on top of clouds.

**P3,L26-27: "... can reach the upper troposphere and modify the PV, consequently allowing for exchange between tropospheric and stratospheric air..." -> Note, however, that WCB air masses do not necessarily enter the stratosphere; the diabatic heating during the ascent is associated with mid-tropospheric PV changes, and the WCB is also able to modify the upper-level PV, but further diabatic and/or diffusive processes are needed that the air masses cross the tropopause.**

We are aware that the ascent of WCB air masses does not necessarily end in the stratosphere. This is a rather uncommon process and additional processes are required to allow for STE (Madonna et al., 2014). As our study shows and as has been hypothesized in a previous study (Kunkel et al., 2016), there is a potential of air masses undergoing STE above the WCB outflow due to turbulent processes. In the current study the turbulence has been observed on top of the WCB outflow (Figure 2a) and the ECMWF model also predicts ice clouds up to the tropopause as it is the case in our idealized experiments. In Kunkel et al. (2016) it has been shown that the turbulent kinetic energy is enhanced on top of the ice clouds. Other numerical studies showed that clouds can substantially alter the PV structure at the tropopause, consequently allowing for STE (e.g., Gray, 2006, Lamarque and Hess, 1994, Spreitzer et al., 2019).

**P4.L31+33: "took place" & "are to examine"; the tense is switching from past to present; please make this consistent (not only at this place).**

Thanks for the hint, we checked the manuscript for further inconsistencies.

**P5,L33-34: "had the goal to study the abundance of trace species in the extratropical tropopause region in relation to the occurrence of enhanced values of static stability in the lower stratosphere and to potential STE." -> Please rephrase in a clearer way; as a suggestion: ".. in the extratropical tropopause region and how they are influenced by the enhanced static stability ... and potential STE. Such conditions were found by Kunkel et al. (2016) to occur in the ridges of extratropical baroclinic waves. Therefore,…"**

We changed this sentences to:

*"The goal of research flight RF 07 on 28.09.2017 was to study the abundance of trace species in the extratropical tropopause region. In particular, in accordance to the major goals of the WISE mission, the focus of this flight was on whether the trace species show specific signs of recent STE in regions of enhanced values of static stability in the lower stratosphere. Furthermore, the design of this flight was chosen such that the predictions of the idealized simulations of Kunkel et al. (2016) could be supported by observations."*

**P6,L9: Why is a slightly degraded horizontal grid (0.125 deg) for the trajectory calculation compared to the other analysis (0.07 deg)?**

The use of a slightly degraded grid was related to technical issues with the memory allocation why we had to choose a slightly lower resolution for the trajectory calculations. We note here that the differences between these two horizontal resolutions are almost negligible for our purposes.

**P7,L7: The horizontal resolution of the COSMO output is 0.4 deg; in contrast, it is 0.125 deg for ECMWF, i.e. it is higher for ECMWF than for COSMO. Is this correct? How do the vertical spacing of ECMWF and COSMO compare in the UTLS?**

This is correct, ECMWF has a finer horizontal grid spacing, but COSMO the finer vertical grid spacing with 110 m in the UTLS (P7, L7) compared to the ECMWF model with about 300 m in the UTLS (P6, L10).

**Section: 3.1: Would it be possible to have one figure (or figure panel) where all flight legs are labeled? While reading this section it was difficult to immediately know where the flight legs are. For instance, it would help to locate the flight legs of figure 3 more easily.**

Figure 2: Revised Figure 2a, now highlighting FL380 and FL420.

[Figure]

We revised Figure 2a and mark the two flight legs which most important for our study as shown above. The arrows point in the direction of the flight on the respective legs.

**P10,L23-: Here, a model deficiency is discussed, namely the too high values of the Richardson number near some regions around the tropopause. This discussion of a model deficiency somewhat interrupts the main storyline, and hence I wonder whether it should better be discussed in section 2.2 where ECMWF data are introduced? Furthermore, the term 'in some regions' is rather unspecific, and immediately lets the reader ask where these regions are.**

Since the discussion of the Ri-discrepancy is very specific and related to the details of Fig. 3, we decided to leave this at the position in the text. Otherwise a reader would have to go backward and forward in the manuscript, when discussing this later.

Please note that we change *"some regions"* to *"regions around the tropopause where the vertical shear of the horizontal wind is large"*.

**P10,L34-35: "Thus, the model forecast underestimates the strength of the inversion, most potentially due to defincies in representing the gravity wave in this region." -> First, note the spelling error! Then, how sure are you that this is indeed a gravity wave? Then, the underestimation of the strength of the inversion is attributed to the effect of the gravity waves, i.e. because they are not well enough captured by the model. How do you know that this underestimation is not because of a limited vertical (and horizontal) resolution of the model?**

Thanks for pointing out the spelling error.  A limited grid spacing indeed would result in an underestimate of the inversion strengths and we suspect that this is mainly related to the vertical grid spacing. However, the occurrence of gravity waves under these meteorological conditions is shown in e.g. Kunkel et al. (2014, 2016) and confirmed by the GLORIA observations (Fig. 3b).

The gravity wave is not well captured in the ECMWF model due to the limited vertical and horizontal resolution of the model.

We change the text accordingly:

*"Thus, the model forecast underestimates the strength of the inversion, potentially related to deficiencies in representing the gravity wave in this region due the still relatively coarse grid spacing in the UTLS."*

**P13,L2-3: "In general, at the tropopause the CO−N2O correlation starts with larger CO and larger N2O mixing ratios at potential temperatures typical for the extratropical tropopause" > 'larger' refers to a comparison; but to what is it compared? Of course, I see the point, but I think it is not perfectly clear. Furthermore, I wonder whether it is correct to say that a correlation starts at a larger N2O and CO values. It sounds a little strange to me!**

We change this sentence to:

*"In general, at the tropopause the CO-$N_2$O correlation shows almost tropospheric mixing ratios of CO (~90 $ppb_v$) and N2O (~331 $ppb_v$) at potential temperature levels typical for the extratropical tropopause in the current case (~335 K)."*

**P16,L12: What does "at some time" mean? Or, stated otherwise: How long are the backward trajectories? Possibly, I simply missed this piece of information, and if not: Please add it!**

The backward trajectories start at 01 UTC on 28.09.2019 and end at 11 UTC on 29.09.2019. With "some time" we refer to this time interval. We also note that there were wrong start and end dates provided in the text (P16, L14) which have been corrected accordingly and now agree with the times given in the caption of Figure 7. We rephrase this sentence on P16, L12 to:

*"For this we filter all trajectories to find those trajectories which cross the dynamic tropopause and which encounter Richardson numbers smaller than 1 at any point in time during the period of the trajectory calculation."*

**P16,L23.24: " Starting from this region the trajectories strongly decelerate in a region of alternating horizontal divergence. During this time the trajectories cross back and forth over the dynamic tropopause." -> How do you see in figure 7 that the trajectories are decelerating? How do you interpret the alternating horizontal divergence? Is the divergence pattern due to the gravity wave?**

The deceleration is not shown directly, but can be indirectly inferred from Fig 7c where the time around 15 UTC is shown. Trajectories have been calculated between 01 UTC on 28.09.2019 and 11 UTC on 29.09.2019 covering a time range of 34 hours. Figure 7c shows the time relative to 15 UTC on 28.09.2019 which is 14 hours after trajectory start but 20 h before the trajectory end time (thinking in physical time direction). However, the distance covered in the 14 hours before 15 UTC is much larger than the distance covered in the 20 hours after 15 UTC, thus the trajectories must have decelerated. The divergence is most probably related to the gravity wave.

**P16,L26-27: " The motion back and forth across the chosen PV value for the dynamic tropopause becomes also evident from the PV along the trajectories (Figure 8a)." -> I am not sure whether I see this crossing back and forth over the dynamic tropopause in figure 8. What I see is that both sides (PV smaller and larger than 2 PVU) are 'covered' by the trajectories, but no further details.**

We admit that the crossing of individual trajectories is not evident in Figure 8a. However, there is also no coherent ensemble of trajectories which crosses the tropopause only once and then stays in the stratosphere. Analyzing the PV values along the trajectories also provided the information that many trajectories cross the dynamic tropopause several times but not as a coherent ensemble oscillating around the dynamic tropopause..

We change the sentence accordingly:

*"The analysis of the PV along the potential TST trajectories shows that these trajectories rather oscillate around the dynamic tropopause than cross the tropopause as coherent ensemble once and then reside in the lower stratosphere. This is somewhat evident in Figure 8a where only those trajectories are shown which at the physical start time of the trajectories have PV values smaller and at the last physical time step  PV values larger than the respective dynamic tropopause PV value (here 2 pvu)."*

**Figure 7: In the text it is written that a transition (although not a smooth one) can be seen in the PV (panel b). This is difficult to see in my print out. It might also be helpful to have the relevant flight legs added to the figure; otherwise, it is a little 'difficult' to relate the trajectories to the measurements.**

We altered the color scale of Figure 7b to improve the  visualization of this behavior. Regarding the flight legs, the caption states of Figure 7 states: "Trajectories crossing the 2 pvu isosurface and encountering a dynamic instability which cross the flight track between 14:36 –14:48 UTC." Thus, these are the trajectories which cross the flight track at FL380 between the respective time. We will add the information of the flight leg to the caption and hope that along with the new Figure 2a this helps to bring together all necessary information.

**Figure 8: Why does the scale go up to 10 PVU in panel a)? In the same line, would it be possibly to adjust the scale in panel d)? Here, Ellrod and Knapp's TI index is shown as an additional turbulence indicator? Earlier in the text, only Richardson number was considered? I think it would be nice to be consistent throughout the manuscript, i.e. either to discuss only one or both indices. Finally, would it make sense to zoom in into a shorter time period around 15 UTC? For example, from -8 h to + 8 h.**

We change the scale to go up only to 5 PVU in panel a) and to cover only the range of 330 K to 340 K in panel d).
We tend to keep the TI since the TI increases due to both horizontal deformation and/or vertical shear of the horizontal wind. If the TI is large this means that either one of these or both processes must be at work. If the TI is large and Ri is low, we can conclude that the vertical shear is large and causing the turbulence. Thus, this is another confirmation of our hypothesis that the mixing processes is caused by the strong shear of the wind.
In the revised manuscript we shorten the time period shown in Figure 8 and make it symmetric around 15 UTC. We now show the 14 hours prior and after 15 UTC to still demonstrate that the time around 15 UTC is rather unique.

**P19,L3: Where does the number 69048 come from? Is this the starting frequency of the trajectories times the duration of the time period?**

This is the number of trajectories started during the time period between 14:20 – 14:36 UTC.

**P19,L14-23: Here, it is discussed whether the STE (and in particular TST) follows the classical meaning of TST. What is 'the classical meaning of TST'?**

**The term is unclear! Actually, I wonder whether this whole discussion about 'classical' or 'not classical' is necessary? If the authors would like to keep it, a more detailed discussion about the exact meaning of this term has to be included, and it has to be made more clear why it is relevant for the study.**

We hope that we could clarify this issue also brought up by reviewer #01 and refer here to the answer given to reviewer #01. We changed the text here accordingly:

*"However, based on the trajectory analysis it is difficult to estimate whether STE and in particular TST occurs in a way that air parcels cross the dynamic tropopause only once from the troposphere into the stratosphere and stay there afterwards. We also performed longer trajectory calculations which, however, also provide no further evidence of TST trajectories which then stay in the stratosphere. We rather find trajectories which, based on PV, alternate back and forth between troposphere and stratosphere and which encounter low Richardson numbers along their paths. One potential reason why no TST trajectories are found which stay in the stratosphere is that the model fails to correctly resolve the process. "*

**Figure 9 and the corresponding text: This figure shows the maximum static stability in the idealized baroclinic life cycle experiment BRTC LC1. I have several questions with respect to this figure: (i) How robust is the maximum static stability? (ii) Are only STT and TST trajectories in the ridges of a baroclinic wave included? I guess that this is not the case, but if so: It distracts the reader from the main story, which is about STE at exactly these locations.**

Referring to (i): Several studies used the maximum static stability to identify the TIL (e.g., Erler and Wirth, 2011; Gettelman and Wang, 2015; Kaluza et al., 2019). Following Erler and Wirth, 2011 as well as Kunkel et al., 2014, 2016 we use the maximum static stability to trace the appearance of the TIL in the baroclinic life cycles which start from a state with no enhanced static stability but with a stratospheric background value of around $4.0 \times 10^{-4}$ s$^{-2}$. To avoid to introduce spuriously high values we trace the maximum of the static stability which has been averaged over the first kilometer above the first local lapse rate tropopause. For this and based on previous analysis we think that $N^2_{max}$ is a relatively robust measure of the evolution of the enhanced static stability in the life cycle experiments.

Referring to (ii): In this figure we show all TST and STT trajectories which makes from our point of view an even stronger point by not filtering by region. This filtering is done in the consequent figures. The point here is to show that there is a temporal coincidence between the $N^2$ enhancement, initiated by the updraft in the troposphere, i.e., the WCB, and the slightly delayed STE which is in line with the point made before that processes at the top of the WCB outflow can lead to STE. In the following figures we show that there is also a spatial coincidence first in the horizontal (Figure 10) and also in the vertical (Figure 11) between TST and $N^2$ enhancement in the ridge of the baroclinic wave.

**P21,L3-4: "Thus, there is almost a temporal coincidence between the start of the enhancement of static stability and the first occurrence of STE." -> The statement is fine, but it repeats essentially the first sentence of the paragraph. Hence, it is somewhat repetitive!**

We remove this sentence in the revised manuscript.

**Figure 10: Where are these structures relative to the ridge and trough of the baroclinic wave?**

In a revised Figure 10 (panels a and b) we include isolines of potential temperature for 315K and 325 K on the level of the dynamic tropopause to visualize the ridge and trough.

**P22,L4-5: "In contrast to TST and although a temporal coincidence is also evident for N2 enhancement and occurrence of STT, no spatial co-occurrence is evident for STT in regions of enhanced N2 (Figures 10c,d)" -> Rephrase in a clearer way? How do you infer that STT does not co-occur with enhanced N^2? Is this statement based only on the blue area in figure 10? Wouldn't we need a stratospheric tracer to make such a statement?**

We also have a stratospheric tracer but do not show this tracer in the manuscript. For completeness, we show the respective figures here. Please note that the first STT is diagnosed a few hours later than the first TST (see Figure 9). At 60 h after model start, there is also no stratospheric tracer evident at the first tropospheric level below the dynamical tropopause:

Figure 3: Stratospheric tracer on the first tropospheric model level below the dynamic tropopause. Black contour lines show values of static stability of $N^2 = 5.5 \times 10^{-4}$ s$^{-2}$.

[Figure]

[Figure]

As well as there is a spatial overlap between the red areas of F10c,d with the tropospheric tracer, there is a spatial overlap between the blue areas and the stratospheric tracer.
We rephrase the respective sentence to:

*"In contrast to TST and although a temporal coincidence is also evident for $N^2$ enhancement and occurrence of STT, no spatial co-occurrence is evident for STT in regions of enhanced $N^2$ (Figures 10c,d and also based on the analysis of the stratospheric tracer which is not shown explicitly here)."*

**P22,L8: "just above a region of ice cloud occurrence" -> What is the relevance of these ice clouds?**

From Kunkel et al. (2016) it is known that these ice clouds increase the static stability in the lower stratosphere through radiative cooling. More so, these ice clouds show the location of the WCB and most importantly that the WCB reached up to the tropopause. Since in our observational case the clouds also reach up to the tropopause, this gives us more confidence that the idealized experiment strongly resembles the observations.
Regarding the process, on top of the ice clouds usually a radiative cooling is evident which can further enhance the static stability in the lower stratosphere. Also on top of the ice clouds there could be enhanced turbulence due to buoyant heat fluxes caused by strong gradients in total water. This is also addressed in Figures 14b,c.

**P22,L9: "wave pattern related to a propagating inertia gravity wave" -> The gravity wave seems to be rather important for the mixing across the tropopause? It is not completely clear to me where this gravity wave originates from? Further, it is written that the wave propagates? But in which direction? It seems to me, based on the vertical cross sections, that the wave does not really propagate in the vertical direction. Instead, could it be that the gravity wave actually propagates along the troposphere-stratosphere interface, i.e. along the tropopause? A more refined analysis would be very helpful, given that the wave patterns is mentioned at several places in the manuscript.**

The gravity wave is probably emitted from the upper tropospheric jet-front system, just as described for instance by Plougonven and Snyder (2007) or Wei and Zhang (2014). Such gravity waves, so called inertia gravity waves, tend to propagate more in the horizontal direction than mountain waves which mainly propagate vertically. From theoretical and idealized model studies it is known that these waves propagate in close vicinity to the tropopause where they can break or at least dissipate (e.g., Bühler and McIntyre, 2005; Plougonven and Snyder, 2005; Plougonven and Zhang, 2014).

However, from our observations we can not infer much about the wave, since we crossed it only once but do not know exactly how we crossed it in a geometric sense. And although the ECMWF model shows signs of the gravity wave, for instance in the divergence of the horizontal wind, we know from our analysis that the model does not capture this structure correctly as we see from the potential temperature along the flight track (Figure 4).

**P26,L8-9: "A common feature of the two sets of trajectories is that in both cases the potential temperature values hardly change in the six hours; thus, the TST occurs quasi-isentropically." -> How do you see this in the figure? Or is this statement based on a quantitative analysis of the trajectories? I think a more detailed discussion of the quasi-isentropic transport is necessary, in particular because this is one of the key words in the article's title.**

A detailed analysis (not shown) of the trajectories in Figure 8d shows that potential temperature of these trajectories changes about 1 K around the time of the TST but is otherwise almost constant. Also the trajectories in our idealized experiments hardly change their potential temperature, although slightly more than those shown in Figure 8. From Figure 12 this is evident by the almost constant color code along the individual dots of the individual trajectories. Further analysis showed that the potential temperature along these trajectories changes between 1.0-2.5 K over the course of the six hours.

**References:**

Bühler, O. and McIntyre, M. E.: Wave capture and wave–vortex duality, J. Fluid Mech., 534, 67–95, 2005.

Erler, A. R. and Wirth, V.: The Static Stability of the Tropopause Region in Adiabatic Baroclinic Life Cycle Experiments, J. Atmos. Sci., 68(6), 1178–1193, doi:10.1175/2010JAS3694.1, 2011.

Gray, S. L.: Mechanisms of midlatitude cross-tropopause transport using a potential vorticity budget approach, J. Geophys. Res., 111, D17113, doi:10.1029/2005JD006259, 2006.

Kunkel, D., Hoor, P., and Wirth, V.: Can inertia-gravity waves persistently alter the tropopause inversion layer?, Geophysical Research Letters, 41, n/a–n/a, https://doi.org/10.1002/2014GL061970, 2014.

Kunkel, D., Hoor, P., and Wirth, V.: The tropopause inversion layer in baroclinic life-cycle experiments: the role of diabatic processes, Atmospheric Chemistry and Physics, 16, 541–560, https://doi.org/10.5194/acp-16-541-2016, 2016.

Lamarque, J.-F. and Hess, P. G.: Cross-Tropopause Mass Exchange and Potential Vorticity Budget in a Simulated Tropopause Folding, Journal of the Atmospheric Sciences, 51, 2246–2269, https://doi.org/10.1175/1520-0469(1994)051<2246:CTMEAP>2.0.CO;2, 1994.

Madonna, E., Wernli, H., Joos, H. and Martius, O.: Warm Conveyor Belts in the ERA-Interim Dataset (1979–2010). Part I: Climatology and Potential Vorticity Evolution, J. Clim., 27(1), 3–26, doi:10.1175/JCLI-D-12-00720.1, 2014.

Plougonven, R. and Snyder, C.: Gravity waves excited by jets: Propagation versus generation, Geophys. Res. Lett., 32(18), L18802, doi:10.1029/2005GL023730, 2005.

Plougonven, R. and Snyder, C.: Inertia–Gravity Waves Spontaneously Generated by Jets and Fronts. Part I: Different Baroclinic Life Cycles, J. Atmos. Sci., 64, 2502–2520, doi:10.1175/JAS3953.1, 2007.

Plougonven, R. and Zhang, F.: Internal gravity waves from atmospheric jets and fronts, Rev. Geophys., 52, n/a-n/a, doi:10.1002/2012RG000419, 2014.

Spreitzer, E., R. Attinger, M. Boettcher, R. Forbes, H. Wernli, and H. Joos, 2019: Modification of Potential Vorticity near the Tropopause by Nonconservative Processes in the ECMWF Model. J. Atmos. Sci., 76, 1709–1726, https://doi.org/10.1175/JAS-D-18-0295.1

Wei, J. and Zhang, F.: Mesoscale Gravity Waves in Moist Baroclinic Jet–Front Systems, J. Atmos. Sci., 71(3), 929–952, doi:10.1175/JAS-D-13-0171.1, 2014.

Zhang, F., Wei, J., Zhang, M., Bowman, K. P., Pan, L. L., Atlas, E., and Wofsy, S. C.: Aircraft measurements of gravity waves in the upper troposphere and lower stratosphere during the START08 field experiment, Atmos. Chem. Phys., 15, 7667–7684, https://doi.org/10.5194/acp-15-7667-2015, 2015.

---

## Author Response (AR2)

**General remarks on the second revised version of Kunkel et al., 2019: Evidence of small-scale quasi-isentropic mixing in ridges of extra-tropical baroclinic waves**

We wanted to thank the handling editor Peter Haynes as well as the two reviewers for their work with our manuscript. During the revision we changed minor parts of the text of the manuscript which is highlighted here:

- P1L1: extra-tropical → extratropical

- P3L23: Zhang et al., (2015a) → (Zhang et al., 2015a)

- P3L23-24: Reference (Zierl and Wirth, 1997) placed at the end of the sentence.

- P13L1: simultaneous → simultaneously

- P15L1: removed (PSD), since it is not further used.

- We furthermore added a note in the acknowledgement and thank the two reviewers for their valuable comments.

- In the reference section, we added the journal title to the reference of Butchart, 2014 and removed several n/a page references.

**Reply to referee comment #01, report #02**

We will answer to all comments of reviewer #01 below point by point. Referee comments are given in bold, answers in standard, and changes to the manuscript in italic font.

**The authors have properly addressed all my comments on the original manuscript. I believe the paper is suitable for publication subject to a handful of very small technical corrections listed below.**

We appreciate the careful reading of our manuscript and change our manuscript as suggested.

- **P4 L8-9 "Early suggestion were" --> Early suggestions were**
  Changed.

- **P9 L16 "indicative for" --> indicative of**
  Changed and also changed in L09 on the same page.

- **P10 L25-25 "larger values than from the…" --> larger values than that from the…**
  Changed accordingly.

- **P16 L32. In the current version of Fig 8a it does not look like all the trajectories end at PV >2. This got me confused for a moment until I realized that the time axis is now shortened compared to the original version of the figure.**

We added a sentence to the figure caption stating:
*"Note that the time period shown here is shorter than the total time period used for the trajectory analysis."*

- **P19 L2 "in the between" --> In between**
  Changed accordingly.

[revised manuscript text omitted]